# Using Probabilistic Machine Learning to Better Model Temporal Patterns in Parameterizations: a case study with the Lorenz 96 model.

Raghul Parthipan[1,2], Hannah M. Christensen[3], J. Scott Hosking[2,4], and Damon J. Wischik[1]

[1]Department of Computer Science, University of Cambridge, Cambridge, UK
[2]British Antarctic Survey, Cambridge, UK
[3]Department of Physics, University of Oxford, Oxford, UK
[4]The Alan Turing Institute, London, UK

**Correspondence:** Raghul Parthipan (rp542@cam.ac.uk)

**Abstract.** The modelling of small-scale processes is a major source of error in weather and climate models, hindering the accuracy of low-cost models which must approximate such processes through parameterization. Red noise is essential to many operational parameterization schemes, helping model temporal correlations. We show how to build on the successes of red noise by combining the known benefits of stochasticity with machine learning. This is done using a recurrent neural network within a probabilistic framework (L96-RNN). Our model is competitive and often superior to both a bespoke baseline and an existing probabilistic machine learning approach (GAN) when applied to the Lorenz 96 atmospheric simulation. This is due to its superior ability to model temporal patterns compared to standard first-order autoregressive schemes. It also generalises to unseen scenarios. We evaluate across a number of metrics from the literature, and also discuss the benefits of using the probabilistic metric of hold-out likelihood.

## 1 Introduction

A major source of inaccuracies in climate models is due to 'unresolved' processes. These occur at scales smaller than the resolution of the climate model ('sub-grid' scales) but still have key effects on the overall climate. In fact, most of the inter-model spread in how much global surface temperatures increase after $CO_2$ concentrations double is due to the representation of clouds (Schneider et al., 2017; Zelinka et al., 2020). The typical approach to deal with the problem of unresolved processes has been to model the *effects* of these unresolved processes as a function of the resolved ones. This is known as 'parameterization'.

Red noise is a key feature in many parameterization schemes (Christensen et al., 2015; Johnson et al., 2019; Molteni et al., 2011; Palmer et al., 2009; Skamarock et al., 2019; Walters et al., 2019). It is characterized by a power spectral density which is inversely proportional to the square of the frequency. It is the type of noise produced by a random walk, where independent offsets are added to each step to generate the next step. This leads to strong autocorrelations in the time series. It contrasts a white noise process where each step is independent. Hasselmann (1976) developed the theory underpinning the importance of red noise for parameterization, showing that the coupling of processes with different time-scales leads to red noise signals in the longer time-scale process, analogous to what takes place in Brownian motion. In parameterization, the resolved processes

typically have far longer time-scales than the unresolved ones, motivating the inclusion of red noise in these schemes. The usefulness of tracking the history of the system for parameterization follows from the importance of red noise.

We use probabilistic machine learning to propose a data-driven successor to red noise in parameterization. The theory explaining the prevalence of red noise in the climate was developed in an idealized model, within the framework of physics-based differential equations. Such approaches may be intractable outside of the idealized case. Recently, there has been much work looking at uncovering relationships from *data* instead (Arcomano et al., 2022; Beucler et al., 2020, 2021; Bolton and Zanna, 2019; Brenowitz and Bretherton, 2018, 2019; Chattopadhyay et al., 2020a; Gagne et al., 2020; Gentine et al., 2018; Krasnopol-
sky et al., 2013; O'Gorman and Dwyer, 2018; Rasp et al., 2018; Vlachas et al., 2022; Yuval and O'Gorman, 2020; Yuval et al., 2021). We develop a Recurrent Neural Network in a probabilistic framework, combining the benefits of stochasticity with the ability to learn temporal patterns in more flexible ways than permitted by red noise. We use the Lorenz 96 model (Lorenz, 1996), henceforth the L96, as a proof-of-concept. Our model is competitive with, and often outperforms, a bespoke baseline. It also generalises to unseen scenarios as shown in Section 4.3.

## 1.1   Numerical Models

Numerical models represent the state of the Earth system at time $t$ by a state vector $\mathbf{X}_t$. The components of $\mathbf{X}_t$ may include, for example, the mean temperature and humidity at time $t$ in every cell of a grid that covers the Earth. The goal is to parameterize the effects of unresolved processes in such a way that the model can reproduce the evolution of this finite-dimensional representation of the state of the real Earth system. A simple way to model the evolution would be

$$X_{k,t+1} = X_{k,t} + f_k(\mathbf{X}_t) \tag{1}$$

where $X_{k,t}$ is the value that the state variable $X$ takes at the spatial coordinate $k$ and time point $t$ and $X_{k,t} \in \mathbb{R}^d$, $\mathbf{X}_t \in \mathbb{R}^{dK}$, and $f$ is a function for the updating process.

## 1.2   Stochastic Schemes

Stochasticity can be included using the following form

$$X_{k,t+1} = X_{k,t} + f_k(\mathbf{X}_t, H_{k,t+1}) \tag{2}$$
$$H_{k,t+1} = \beta(H_{k,t}, z_{k,t+1}) \tag{3}$$
$$z_{k,t} \sim \mathcal{N}(0, I) \tag{4}$$

where hidden variables (discussed below) created by the modeller are denoted $H_{k,t} \in \mathbb{R}^{dim(H)}$ and are tracked through time,
$\beta$ is a function for updating these, and $z_{k,t} \in \mathbb{R}^{dim(z)}$ is a source of stochasticity.

   Introducing stochasticity through (2) improved on models of the form (1). Results included better ensemble forecasts (Buizza et al., 1999; Leutbecher et al., 2017; Palmer, 2012), and improvements to climate variability (Christensen et al., 2017) and mean climate statistics (Berner et al., 2012). The motivation for using stochasticity comes from the understanding that the effects of

the unresolved (sub-grid) processes cannot be effectively predicted as a deterministic function of the resolved ones due to a lack of scale separation between them. Stochasticity allows us to capture our uncertainty about those aspects of the unresolved processes which may affect the resolved outcomes.

## 1.3 Hidden Variables and Red Noise

Hidden variables — $H_{k,t}$ in (2) and (3) — are defined here as being variables separate to the observed state, $X_{k,t}$, which if tracked help better model $X_{k,t}$. Using them is key to numerical weather and climate models using stochastic parameterizations, allowing temporal correlations to be better modelled. Red noise results when the hidden variables evolve with a first-order autoregressive (AR1) process such as

$$H_{k,t+1} = \phi H_{k,t} + \sigma(1 - \phi^2)^{1/2} z_{k,t+1} \tag{5}$$

where $\phi$ is the lag-one correlation and $\sigma$ is the standard deviation.

One example is the stochastically perturbed parameterization tendencies (SPPT) scheme (Buizza et al., 1999; Palmer et al., 2009) which is frequently employed in forecasting models (Leutbecher et al., 2017; Molteni et al., 2011; Palmer et al., 2009; Sanchez et al., 2016; Skamarock et al., 2019; Stockdale et al., 2011). Here, the AR1 process results in far better weather and climate forecasting skill than a simple white noise model, with good modelling of regime behaviour requiring correlated noise (Christensen et al., 2015; Dawson and Palmer, 2015).

There is no intrinsic reason why an AR1 process is the best way to deal with these correlations. It is simply a modelling choice.

## 1.4 Machine Learning for Parameterization

Learning the parameters of a climate model, either the simple form (1) or the general form (2)–(4), requires deciding on a parametric form for the functions $f$ and $\beta$. For full-scale climate models, it is difficult to find appropriate functions. The machine learning (ML) approach is to *learn* these from data. Various researchers have proposed ML methods for learning the deterministic (meaning the same output is always returned for a given input) model (1) (Beucler et al., 2020, 2021; Bolton and Zanna, 2019; Brenowitz and Bretherton, 2018, 2019; Gentine et al., 2018; Krasnopolsky et al., 2013; O'Gorman and Dwyer, 2018; Rasp et al., 2018; Yuval and O'Gorman, 2020; Yuval et al., 2021).

Amongst ML-trained stochastic models (Gagne et al., 2020; Guillaumin and Zanna, 2021), various ones with red noise were proposed by Gagne et al. (2020), using Generative Adversarial Networks (GANs) (Goodfellow et al., 2014). Full architectural details are in Gagne et al. (2020) and we will refer to one of their best-performing models (which they call X-sml-r) as the *GAN*. A wider range of generative models can be trained using such an advesarial approach as opposed to maximum likelihood (the standard way to train models in ML, discussed in Section 3) but such methods are notoriously unstable due to the nature of the minimax loss in training (Goodfellow, 2016). Although they used ML to learn $f$ in (2), it was not used to learn $\beta$, which they modelled with an AR1 process.

Recurrent Neural Networks (RNNs) are a popular ML tool for modelling temporally correlated data, eliminating the need for update functions (like that in equation (3)) to be manually specified. RNNs have had great success in the ML literature in a variety of sequence modelling tasks, including text generation (Graves, 2013; Sutskever et al., 2011), machine translation (Sutskever et al., 2014) and music generation (Eck and Schmidhuber, 2002; Mogren, 2016). The state-of-the-art RNNs are long short-term memory (LSTM) networks (Hochreiter and Schmidhuber, 1997) and gated recurrent unit (GRU) networks (Cho et al., 2014). These use gating mechanisms to control information flow, providing major improvements to the vanilla RNN's issue of unstable gradients and therefore making it easier to capture long-range dependencies.

Current parameterization work using ML to model temporal correlations have predominantly used deterministic approaches, including deterministic RNNs and echo state networks (Arcomano et al., 2022; Chattopadhyay et al., 2020a, b; Vlachas et al., 2018, 2020, 2022) and found notable success. A stochastic RNN for ocean modelling was recently presented by Agarwal et al. (2021). Our work differs in how we provide a joint training approach for the deterministic and stochastic parts. All this work with RNNs suggest such ML approaches are effective ways to model temporal dynamics in physical systems. However, the investigation of how to implement these in probabilistic frameworks is limited.

Other off-the-shelf models are not obviously suited for the parameterization task. Transformers and attention-based models (Vaswani et al., 2017) perform well for sequences but require all the previous data to be tracked for each simulation step, providing a computational burden which may increase simulation cost. Random Forests (RFs) have been used for parameterization (O'Gorman and Dwyer, 2018; Yuval and O'Gorman, 2020) and shown to be stable at run-time (due to predicting averages form the training set) but it is not obvious how they would learn and track hidden variables.

### 1.5 Overview

The L96 set-up and baselines are presented in Section 2. Our model is detailed in Section 3, with the experiments and results in Section 4. This is followed by a discussion on the use of 'likelihood' in evaluating probabilistic climate models (Section 5), with our conclusions in Section 6.

## 2 Parameterization in the Lorenz 96

We introduce the L96 model here and then present two L96 parameterization models from the literature which help clarify the above discussion and serve as our baselines.

### 2.1 Lorenz 96 Model

We use the two-tier L96 model, a toy model for atmospheric circulation that is extensively used for stochastic parameterization studies (Arnold et al., 2013; Crommelin and Vanden-Eijnden, 2008; Gagne et al., 2020; Kwasniok, 2012; Rasp, 2020). We use the configuration described in Gagne et al. (2020). It comprises two types of variables: a large, low-frequency variable, $X_k$,

interacting with small, high-frequency variables, $Y_j$. These are dimensionless quantities, evolving as follows:

$$\frac{dX_k}{dt} = \underbrace{-X_{k-1}(X_{k-2} - X_{k+1})}_{\text{advection}} \underbrace{-X_k}_{\text{diffusion}} \underbrace{+F}_{\text{forcing}} \underbrace{-\frac{hc}{b}\sum_{j=J(k-1)+1}^{kJ} Y_j}_{\text{coupling}} \qquad k = 1, ..., K$$

$$\frac{dY_j}{dt} = \underbrace{-cbY_{j+1}(Y_{j+2} - Y_{j-1})}_{\text{advection}} \underbrace{-cY_j}_{\text{diffusion}} \underbrace{-\frac{hc}{b}X_{int[(j-1)/J]+1}}_{\text{coupling}} \qquad j = 1, ..., JK$$

where the variables have cyclic boundary conditions: $X_{k+K} = X_k$ and $Y_{j+JK} = Y_j$. In our experiments, the number of $X$ variables is $K = 8$, and the number of $Y$ variables per $X$ is $J = 32$. The value of the constants are set to $h = 1$, $b = 10$ and $c = 10$. These indicate that the fast variable evolves ten times faster than the slow variable and has one-tenth the amplitude. The chosen parameters follow those used in Arnold et al. (2013), and are such that one model time unit (MTU) is equivalent to five atmospheric days when comparing the error doubling time from the model to that seen in the atmosphere (Lorenz, 1996).

The L96 model is useful for parameterization work as we can consider the $X_k$ to be coarse processes resolved in both low-resolution and high-resolution simulators, whilst the $Y_j$ can be considered as those that can only be resolved in high-resolution, computationally expensive simulators. In this study the coupled set of L96 equations are treated as the 'truth', and the aim is to learn a good model for the evolution of $X$ alone using this 'truth' data. The effects of $Y_j$ must therefore be parameterized. Success would be if the modelled evolution of $X$ matched that from the truth.

The L96 is also useful as it contains separate persistent dynamical regimes, which change with different values of $F$ (Christensen et al., 2015; Lorenz, 2006). The dominant regime exhibits a wave-two pattern around the ring of $X$ variables, whilst the rarer regime exhibits a wave-one type pattern. In the atmosphere, regimes include persistent circulation patterns like the Pacific/North American (PNA) pattern and the North Atlantic Oscillation (NAO).

It is easy to use models with no memory which do well on standard loss metrics but fail to capture this interesting regime behaviour. This is seen in the L96, where including temporally correlated (red) noise improves the statistics describing regime persistence and frequency of occurrence (Christensen et al., 2015) as well as improving weather forecasting skill (Arnold et al., 2013; Palmer, 2012). Temporally correlated noise also improves regime behaviour (frequency of occurrence and persistence) in operational models (Dawson and Palmer, 2015), as well as forecasting skill (Palmer et al., 2009).

## 2.2 Parameterization Models in the Literature

The below models are stochastic. They use AR1 processes to model temporal correlations, but this need not be the case, as we show in Section 3.

### 2.2.1 Stochastic Non-ML Model with Non-ML Hidden Variables

Arnold et al. (2013) propose a model which we refer to as the *polynomial* model (in light of the polynomial in (6)) which is

$$X_{k,t+1} = X_{k,t} + \omega_k(\mathbf{X}_t) - \Delta t \left( aX_{k,t}^3 + bX_{k,t}^2 + cX_{k,t} + d + H_{k,t+1} \right) \tag{6}$$

where $H_{k,t}$ evolves as in (5) with $H_{k,1} = \sigma z_{k,1}$, and where

$$\omega_k(\mathbf{X}_t) = \lambda_k \left( \mathbf{X}_t + \lambda(\mathbf{X}_t)/2 \right) \tag{7}$$

and

$$\lambda_k(\mathbf{X}_t) = \Delta t \left( - X_{k-1,t}(X_{k-2,t} - X_{k+1,t}) - X_{k,t} + F \right) \tag{8}$$

where (7) is the implementation of a second order Runge-Kutta method and $[\lambda(\mathbf{a})]_k = \lambda_k(\mathbf{a})$. $H_{k,t}$ only depends on $H_{k,t-1}$.

### 2.2.2 Stochastic ML Model with Non-ML Hidden Variables

The GAN from Gagne et al. (2020) replaces (6) by

$$X_{k,t+1} = X_{k,t} + \omega_k(\mathbf{X}_t) - \Delta t \, U(X_{k,t}, H_{k,t+1}) \tag{9}$$

where the function $U$ is implemented by a neural network (NN), $\omega$ is defined in (7)–(8), $H_{k,t}$ evolves as in (5) with $\sigma = 1$, and $H_{k,1} = z_{k,1}$.

## 3 Our L96-RNN Model

### 3.1 Stochastic ML Model with ML Hidden Variables

Our model, henceforth denoted the *L96-RNN*, follows the general form in (2)–(4) but splits the hidden variable into two parts: $H_{k,t} = (R_{k,t}, l_{k,t})$. The model is

$$X_{k,t+1} = X_{k,t} + \omega_k(\mathbf{X}_t) - \Delta t \left( g_\theta(X_{k,t}) + R_{k,t+1} \right) \tag{10}$$

$$R_{k,t+1} = b_\theta(l_{k,t+1}) + \sigma z_{k,t+1} \tag{11}$$

$$l_{k,t+1} = s_\theta(l_{k,t}, R_{k,t}) \tag{12}$$

where $g$, $b$ and $s$ are NNs with weights $\theta$, $z_{k,t} \in \mathbb{R}^1$ is exogenous noise as defined in (4), $\omega$ is specified in equation (7) above (implements a second order Runge-Kutta method and expresses physical quantities like advection), and $\theta$ and $\sigma$ are parameters to be learnt. We set $l_{k,1} = 0$ as is typically done with RNNs and $R_{k,1} = \sigma z_{k,1}$. The dimensions of the variables are: $X_{k,t} \in \mathbb{R}^1, \mathbf{X}_t \in \mathbb{R}^K, R_{k,t} \in \mathbb{R}^1, \mathbf{R}_t \in \mathbb{R}^K, l_{k,t} \in \mathbb{R}^8, \mathbf{l}_t \in \mathbb{R}^{8K}$. The dimensions of the NNs are: $g_\theta : \mathbb{R}^1 \to \mathbb{R}^1, b_\theta : \mathbb{R}^8 \to \mathbb{R}^1$

and $s_\theta : \mathbb{R}^9 \to \mathbb{R}^8$. It is structurally based on a Recurrent Neural Network (RNN) as it has hidden variables, the same update procedure is used each step, and $g$, $b$ and $s$ are NNs. In this model, the sub-grid parameterization is split into a deterministic part (parameterized by $g_\theta$) and a stochastic part, $R$, which is parameterized by a RNN ($b_\theta$ and $s_\theta$). The RNN is described by equations (11)-(12) and models the stochastic term at the next time step $R_{k,t+1}$ given the past sequence of $R_{k,1} = r_{k,1}, ..., R_{k,t} = r_{k,t}$, where we use upper-case $R_{k,t}$ to denote the random variable and use lower-case $r_{k,t}$ to denote the values which a random variable $R_{k,t}$ may take. When the random variable being referred to is clear from the context, we drop it and just write $r_{k,1}, ..., r_{k,t}$, for example. The same notation is used for $X$. The notation is only important when talking about the probabilistic perspective of the model and the likelihood. Otherwise, the distinction between the upper- and lower-case variables can be ignored by the reader. The backbone of the L96-RNN is the mapping $s_\theta$ which takes as input $r_{k,t}$ and the hidden state $l_{k,t}$ (which summarises the information in $r_{k,t-1}, ..., r_{k,1}$) and updates the hidden state. The stochastic term is then modelled using a Gaussian distribution parameterized by the output layers of the L96-RNN, $b_\theta$, and $\sigma$. The sampled value, $r_{k,t+1}$, is used in equation (10) to output $X_{k,t+1} = x_{k,t+1}$ and is fed back into the L96-RNN.

The key insight is that our model allows more flexibility than the standard AR1 processes for expressing temporal relationships. This is seen by how equation (10) is identical in form to (6), but the hidden variable is evolved in a more flexible manner than the AR1 process in (5).

Figure 1 shows the mechanism of generation and the NN architectures used to learn the functions. The main architectural details are that $s$ in (12) is implemented using two GRU layers, each composed of four units, and $b$ in (11) is represented with a dense layer of size one. $g$ in (10) is implemented using three fully-connected layers.

Here, as well as in the baselines, the parameterization models are 'spatially local' meaning that the full $\mathbf{X}_t$ vector is not taken in as input when modelling $X_{k,t+1}$ anywhere apart from in $\omega_k(\mathbf{X}_t)$. This is done to mimic parameterizations in operational weather and climate models. For all forecast models, $\Delta t = 0.005$ model time units (MTU).

## 3.2 Training Probability Models Using Likelihood

The L96-RNN is probabilistic and trained by maximizing the likelihood of sequences $\mathbf{x}_t$ in the training data, as is commonly done in the ML literature for RNN models (Bahdanau et al., 2014; Cho et al., 2014; Goodfellow et al., 2016; Sutskever et al., 2011, 2014). The 'likelihood of a sequence' is denoted $\mathrm{Pr}(x_1, x_2, ..., x_n)$ and can be interpreted as the probability assigned to a given sequence of variables $(x_1, x_2, ..., x_n)$ by a given model.

For our L96-RNN, the log-likelihood (the natural logarithm of the likelihood) of a sequence of $\mathbf{x}_t$ is

$$\log \mathrm{Pr}(\mathbf{x}_1, ..., \mathbf{x}_n; \mathbf{x}_0) = \sum_{k=1}^{K} \Big( \log \mathrm{Pr}(r_{k,1}) + \sum_{t=2}^{n} \log \mathrm{Pr}(r_{k,t}|l_{k,t}) \Big) - Kn \log(\Delta t) \tag{13}$$

where the semicolon denotes that $\mathbf{x}_0$ is given, and $r$ and $l$ are deterministic functions of the training sequence $\mathbf{x}_0, ..., \mathbf{x}_n$, derived from equations (10)–(12). Specifically, by rearranging (10) we get

$$r_{k,t} = \frac{x_{k,t+1} - x_{k,t} - \omega_k(\mathbf{x}_t)}{-\Delta t} - g_\theta(x_{k,t}) \tag{14}$$

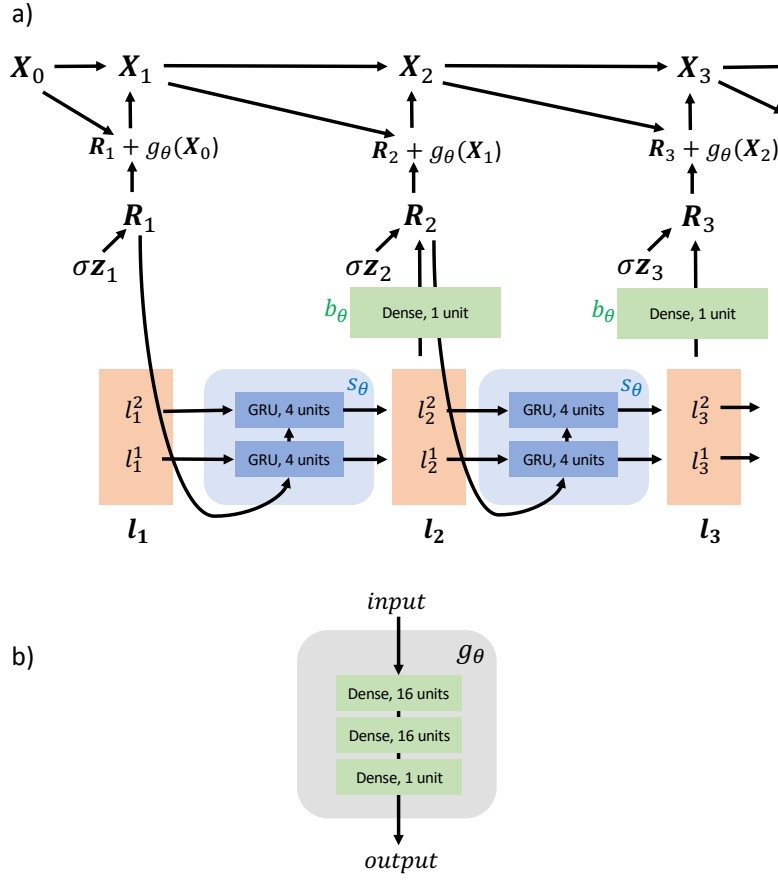

**Figure 1.** (a) RNN graphical model showing how each $\mathbf{X}_t$ is generated. $\mathbf{X}_{t+1}$ is a function of $\mathbf{X}_t$ and $\mathbf{R}_{t+1}$, and $\mathbf{R}_t$ is a function of $\mathbf{l}_t$ and $\mathbf{z}_t$. $\mathbf{z}_t \in \mathbb{R}^K$ is the source of stochasticity. $b_\theta$, $s_\theta$ and $g_\theta$ are NNs. $\mathbf{l}_t$ consists of a stack of $l_t^1$ and $l_t^2$, each $\in \mathbb{R}^4$. The neural networks are stacked $K$ times (not shown in the figure), with each section of the stack being applied to each of the $K$ components of $\mathbf{X_t}$. (b) The architecture of the $g_\theta$ neural network.

and so we can calculate what values $r_{k,t}$ must take in order for the training sequence $\mathbf{x}_0, ..., \mathbf{x}_n$ to be generated. Now given the known sequence $\mathbf{r}_1, ..., \mathbf{r}_n$, and the initial value $\mathbf{l}_1 = \mathbf{0}$, the sequence of $l$ values, $\mathbf{l}_1, ..., \mathbf{l}_n$ can be calculated from equation (12). For example, $l_{k,3} = s_\theta\big(s_\theta(l_{k,1}, r_{k,1}), r_{k,2}\big)$. To reduce clutter in the final equation which we show in this section, we will rewrite equation (14) as

$$r_{k,t} = U_{k,t} - g_\theta(x_{k,t}) \tag{15}$$

where $U_{k,t} = \frac{x_{k,t+1} - x_{k,t} - \omega_k(\mathbf{x}_t)}{-\Delta t}$ and can be understood as the true sub-grid forcing. Now, the explicit expression for the log-likelihood is given by

$$\log \Pr(\mathbf{x}_1, ..., \mathbf{x}_n; \mathbf{x}_0) = \sum_{k=1}^{K} \Bigg( -\log(\sigma\sqrt{2\pi}) - \frac{1}{2\sigma^2}\big(U_{k,0} - g_\theta(x_{k,0})\big)^2$$

$$+ \sum_{t=2}^{n} \Big( -\log(\sigma\sqrt{2\pi}) - \frac{1}{2\sigma^2}\big(U_{k,t} - g_\theta(x_{k,t}) - b_\theta(l_{k,t})\big)^2 \Big) \Bigg) - Kn\log(\Delta t) \qquad (16)$$

The L96-RNN is trained by maximizing this likelihood with respect to the parameters $\theta$ and $\sigma$. The explicit form of the L96-RNN likelihood makes training by maximum likelihood relatively straightforward. The full derivation of the likelihood is in Appendix A.

From equation (16) we can see that both the deterministic part ($g_\theta$) and the stochastic model ($b_\theta$ and $s_\theta$, which is involved via $l_{k,t}$) are trained jointly to minimise the mean-squared-error of the sub-grid forcing prediction. This is unlike the polynomial and GAN models where the deterministic part is fitted first, and then the stochastic part is modelled in a separate step.

### 3.3 RNN Training

Truth data for training was created by running the full L96 model five separate times with the following values of $F$: $(19, 20, 20.5, 21, 21.5)$. Henceforth, we use 'truth' data to refer to data from the full L96 model. We gave our model data from different forcing scenarios so it could learn how perturbations in the forcing may affect the evolution of $\mathbf{X}_t$. The GAN and polynomial were trained in a similar manner to allow for a fair comparison. The truth data was created by solving the two-level L96 equations using a fourth-order Runge-Kutta timestepping scheme and a fixed time step $\Delta t = 0.001$ MTU, with the output saved at every 0.005 MTU. A training set of length 2,500 MTU was assembled from the truth data, consisting of three equal 500 MTU components with $F = (19, 20.5, 21)$ and one 1,000 MTU component with $F = 20$, with a $F = 21.5$ set of length 500 MTU kept as a validation hold-out set. This resulted in a 75:25 training:validation set split (3,999,992 samples: 800,008 samples).

The L96-RNN was trained using truncated back propagation through time on sequences of length 700 time steps for 100 epochs with a batch size of 32 using Adam (Kingma and Ba, 2014), with $\theta$ and $\sigma$ being the learnable parameters. A variable learning rate was used, starting at 0.0001 for the first 70 epochs and decayed to 0.00003 for the remaining 30. A small grid search was conducted over the number of GRU layers (searching over 1 to 3) and the GRU unit sizes (searching between 4 to 16). The model parameters which gave the lowest loss on the validation set were saved.

### 4 Results

This section analyses performance across a range of time-scales. The first results are for $F = 20$. Assessing the model in the training realm allows us to verify if it can replicate the L96 attractor. Later experiments are for $F \geq 28$. For $F = 20$ and $F = 28$, 50,000 MTU long simulations ($\approx 685$ 'atmospheric years') were created for analysis.

## 4.1 Weather Evaluation

The models were evaluated in a weather forecast framework. 745 initial conditions were randomly taken from the truth data and an ensemble of 40 forecasts each lasting 3.5 MTU were generated from each initial condition. Figure 2 shows the spread and error terms for these experiments over time. The error is defined as

$$\text{error}(t) = \sqrt{\frac{1}{M} \sum_{m=1}^{M} \left( X_m^O(t) - \overline{X_m^{\text{sample}}(t)} \right)^2}$$

where there are $M$ different initial conditions, $X_m^O(t)$ is the observed state at time $t$ for the $m^{\text{th}}$ initial condition, and $\overline{X_m^{\text{sample}}(t)}$ is the ensemble mean forecast at time $t$.

The spread is defined as

$$\text{spread}(t) = \sqrt{\frac{1}{M} \sum_{m=1}^{M} \frac{1}{N} \sum_{n=1}^{N} \left( X_{m,n}^{\text{sample}}(t) - \overline{X_m^{\text{sample}}(t)} \right)^2}$$

where $N$ is the number of ensemble members and $X_{m,n}^{\text{sample}}(t)$ is the state of the $n^{\text{th}}$ member at time $t$ for the $m^{\text{th}}$ initial condition.

As noted by Leutbecher and Palmer (2008), for a perfectly reliable forecast (defined as one where $X_{m,n}^{\text{sample}}(t)$ and $X_m^O(t)$ are independent samples from the same distribution), for large $N$ and $M$ the error should be equal to the spread. A smaller error implies an ensemble forecast that better matches observations, and a spread/error ratio close to one implies a reliable forecast. Figure 2 shows that the polynomial and L96-RNN have the smallest errors, but the L96-RNN has the better spread/error ratio until 1.0 MTU. After that, its spread is still slightly better matched to its error compared to the polynomial. The GAN is underdispersive, with the greatest error but smallest spread/error ratio, suggesting it is overconfident in its predictions.

## 4.2 Climate Evaluation

The ability to simulate the climate of the L96 was evaluated using the 50,000 MTU simulations. We use histograms to represent various climate-related distributions. For example, Figure 3a represents each model's marginal distribution of $X_{k,t}$. Since the true continuous distributions for this quantity are unavailable, we represent them using histograms (a discrete estimate). The idea of estimating an underlying continuous distribution's density with a discrete distribution based on a finite set of sampled points is commonplace and known as histogram density estimation (Silverman, 1986; Wilks, 2011). When plotting histograms, an appropriate bin width is necessary. Too few bins will lead to over-smoothing where the underlying distribution's shape is not captured, whilst too many bins will lead to under-smoothing. Both cases lead to poor representations of the underlying distribution. We use the Freedman-Diaconis rule (Freedman and Diaconis, 1981) to determine the bin widths (and therefore number of bins). The rule is based on minizing the mean integrated squared error between the histogram's density estimate and the true density function. The number of bins used are noted in the relevant figure captions.

We can quantitatively measure how well the histogram density estimates from each model match the histogram density estimate from the truth using KL-divergence. The KL divergence has a direct equivalence to log-likelihood: minimizing the

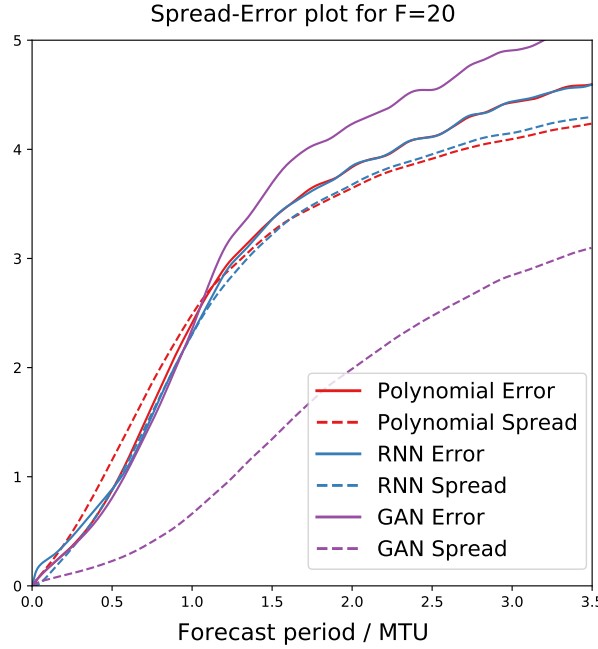

**Figure 2.** Error and spread for weather experiments. We expect a smaller error for a more accurate forecast model. The spread/error ratio would be close to one in a 'perfectly reliable' forecast.

KL-divergence between a true distribution and a model's distribution is equivalent to maximizing the log-likelihood of the true data under the model. We evaluate the KL-divergence between $q(a)$, the distribution described by the histogram of a variable $A$ in the true L96 model, and $p(a)$, the distribution described by the histogram of that variable in a parameterized model, using

$$\mathrm{KL}(q(a)||p(a)) = \sum_a q(a) \log \frac{q(a)}{p(a)}$$

where the sum is over all the discrete values which $A$ takes. The smaller this measure, the better the goodness-of-fit. The calculated KL divergence is obviously dependent on the number of bins used in our histograms. Using number of bins $= 1$ would give a KL divergence of 0, whilst using number of bins $= n$, where $n$ is the number of data points, would give a KL divergence of $\infty$ in this case as the variables under consideration are continuous so their empirical probability density functions will not overlap. Therefore, a bin number which best represents the underlying distribution is important to use. As noted earlier,
we use the Freedman-Diaconis rule for this. Nevertheless, we also provide results in Appendix B for the KL divergences for a range of bin numbers centered on the Freedman-Diaconis suggestions.

Successful reproduction of the L96 climate would result in a model's histogram matching the truth model's. Qualitatively, from Figure 3, the polynomial and L96-RNN are closest to the truth, with the L96-RNN doing slightly better around $X_{k,t} = 9$. Quantitatively, the KL-divergence results (Table 1) confirm that the L96-RNN best matches the truth model here. Results from
250 Appendix B confirm that the stronger L96-RNN performance is also seen over a range of bin numbers.

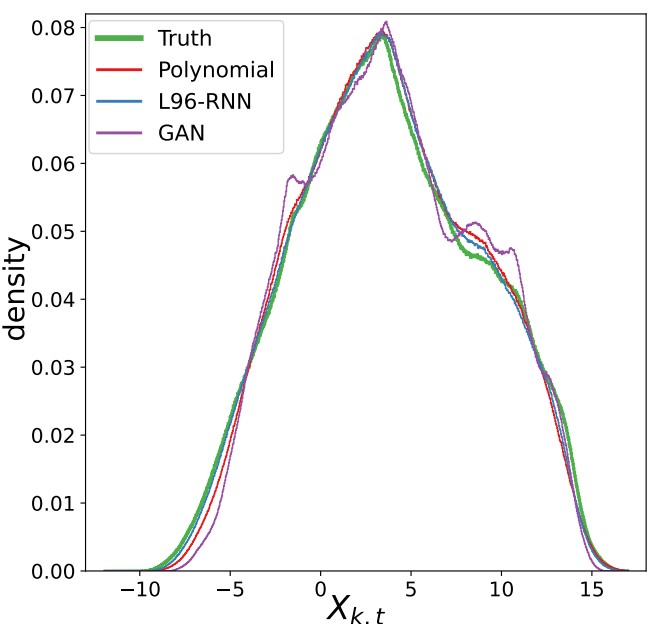

**Figure 3.** Histograms of $X_{k,t}$. The truth histogram is shown in green and **bold**. The closer a model's histogram to the truth, the better it models the probability density of $X_{k,t}$. 827 bins are used for all shown models.

**Table 1.** *KL-divergence (goodness-of-fit) between 1) the truth and 2) the parameterized models, for the distributions shown in the noted Figures. The smaller the KL-divergence, the better the match between the true and modelled distributions. The best model in each case is shown in **bold**.*

| Relevant figure | Polynomial | L96-RNN | GAN |
|:---:|:---:|:---:|:---:|
| 3a | 0.13 | **0.04** | 0.83 |
| 4 | 64 | **32** | 735 |
| 5a | 0.59 | **0.45** | 7.70 |
| 5b | 0.54 | **0.05** | 8.36 |
| 7a | - | **0.1** | 0.5 |
| 7b | - | **0.1** | 1.1 |

The L96 model used here displays two distinct regimes with separate dynamics. We use the approach from Christensen et al. (2015) to examine the regimes. First, the time series are temporally smoothed with a running average over 0.4 MTU to help identify regimes (Stephenson et al., 2004; Straus et al., 2007). The dimensionality of the system is then reduced using Principal Component Analysis, as is often done when studying atmospheric data (Selten and Branstator, 2004; Straus et al., 2007). For the truth time series, the components $PC1$ (Principal Component 1) and $PC2$ (Principal Component 2) are

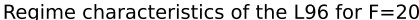

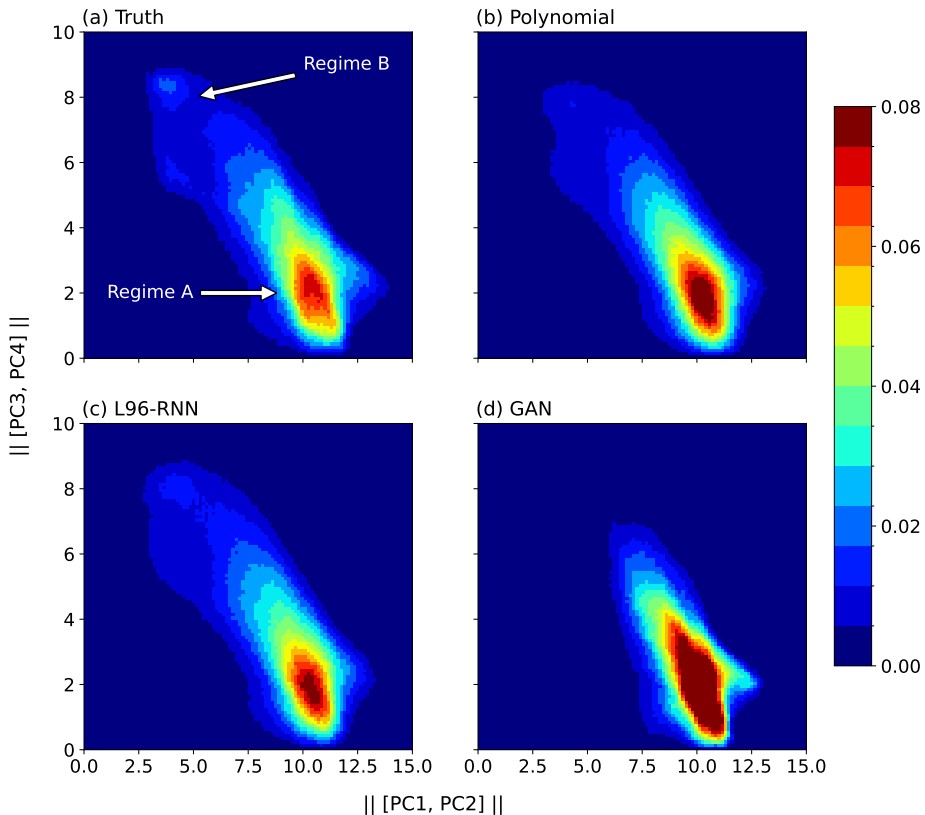

**Figure 4.** Regime characteristics of the L96 for F = 20. 2D histograms show the magnitudes of the projections of $X_{k,t}$ on to the principle components from the truth series. 100 bins are used for this visualisation. When calculating the KL-divergence in Table 1, 350 bins are used.

degenerate, offset in phase by $\pi/2$ radians and represent wavenumber two oscillations. $PC3$ and $PC4$ are also degenerate, offset in phase by $\pi/2$ radians and represent wavenumber one oscillations. All model runs are projected onto the truth model's components. Given the degeneracies, the magnitude of the principal component vectors, $||[PC1, PC2]||$ and $||[PC3, PC4]||$, where $||[PC1, PC2]|| = \sqrt{PC1^2 + PC2^2}$, are computed and histograms of the system are plotted in this space.

The presence of two regimes is apparent in Figure 4a where there is a large maximum corresponding to the major regime, A, but also a smaller peak corresponding to the minor regime, B. The polynomial and GAN fail to capture the two regimes, whereas the L96-RNN is more successful. Comparison is made easier by decomposing the 2D Figure 4 into two, 1D density plots (Figure 5). We can use the KL-divergence to measure how well our models match the truth across all three histograms. Table 1 shows the L96-RNN best matches the truth and so captures the regime characteristics best. There are still evidently

deficiencies though as further density is required to be placed around Regime B.

We desire our parameterized models to correctly capture the temporal behaviour of the L96 system. One way to assess this is by considering the temporal correlations (linear associations) using an autocorrelation plot. The autocorrelation plots of the

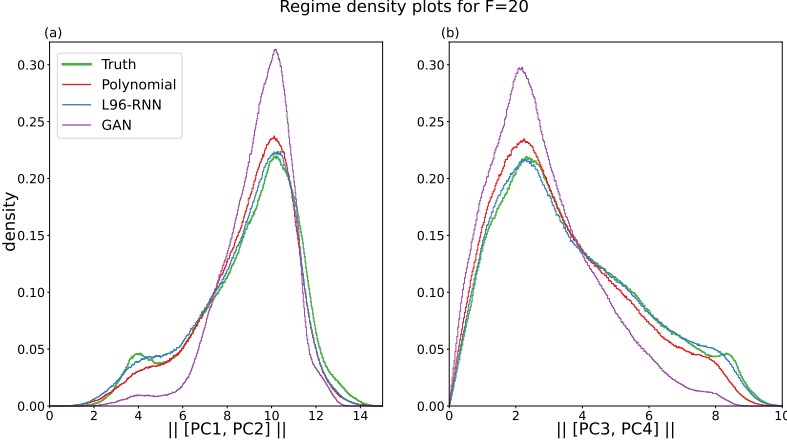
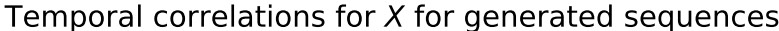

**Figure 5.** Density plots for each regime for $F = 20$. The truth is shown in green and **bold**. (a) Magnitude of $[PC1,PC2]$. 534 bins are used for this. (b) Magnitude of $[PC3,PC4]$. 350 bins are used for this.

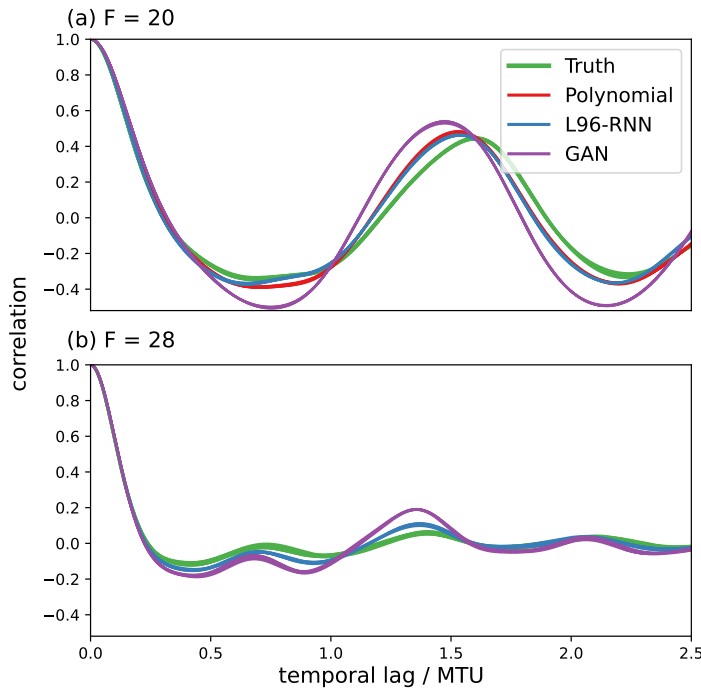

**Figure 6.** Ability of the parameterized models to reproduce the "true" temporal correlations of the $X$ variables in the L96 system (green). The correlations for each of the $K$ components are plotted - the variability is indicated by the line width. (a) is for the sequences generated with F = 20, and (b) is for those with F = 28. The polynomial is not shown for (b) as the simulations are unstable and go to infinity.

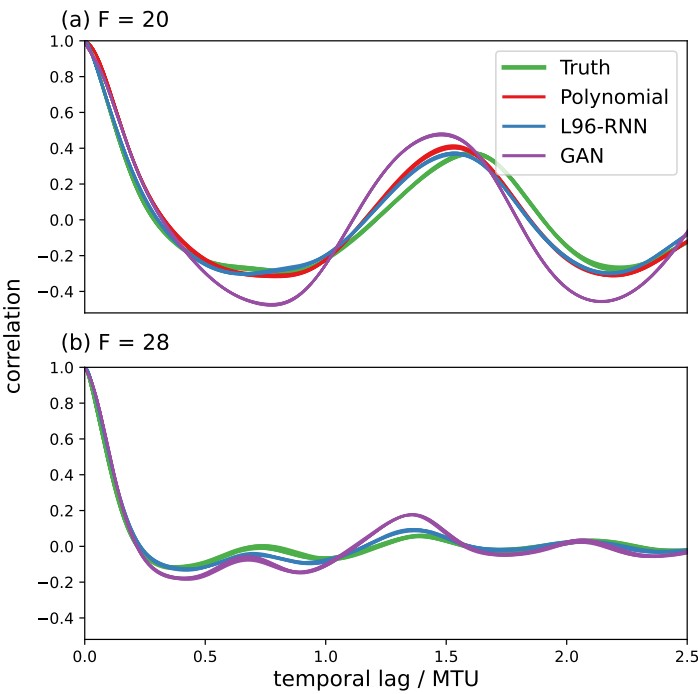

**Figure 7.** Ability of the parameterized models to reproduce the "true" temporal correlations of the sub-grid forcing terms ($U$) in the L96 system (green). The correlations for each of the $K$ components are plotted - the variability is indicated by the line width. (a) is for the sequences generated with F = 20, and (b) is for those with F = 28. The polynomial is not shown for (b) as the simulations are unstable and go to infinity.

generated $X$ variables are shown in Figure 6, and those of the corresponding sub-grid forcing terms, $U$, are shown in Figure 7, where $U_{k,t} = \frac{x_{k,t+1}-x_{k,t}-\omega_k(\mathbf{x}_t)}{-\Delta t}$. In Figure 6(a) we see that for F = 20, all the parameterized models perform well, but the

L96-RNN best captures the trough behaviour at the 0.7 temporal lag. In Figure 7(a), for F = 20, the L96-RNN and polynomial perform the best.

### 4.3   Generalisation Experiments

We set the L96 forcing to $F = 28, 32, 35$ and $40$, and examine how the models can capture changes due to varying external forcings. These $F$ values are notably different to those in the training data so allow us to test generalisation. For example, the

change from $F = 21.5$ (validation set) to $F = 28$ results in the following changes to the regime structure: the centroid location of the rarer regime shifts to higher values of $||[PC1, PC2]||$ and $||[PC3, PC4]||$ (Figure 8), and the proportion of time spent in the rarer regime increases from $38\%$ to $50\%$. The range of $X_{k,t}$ also increases from $[-12, 19]$ to $[-24, 29]$. The differences in wave behaviour between regimes means the above changes result in different system dynamics.

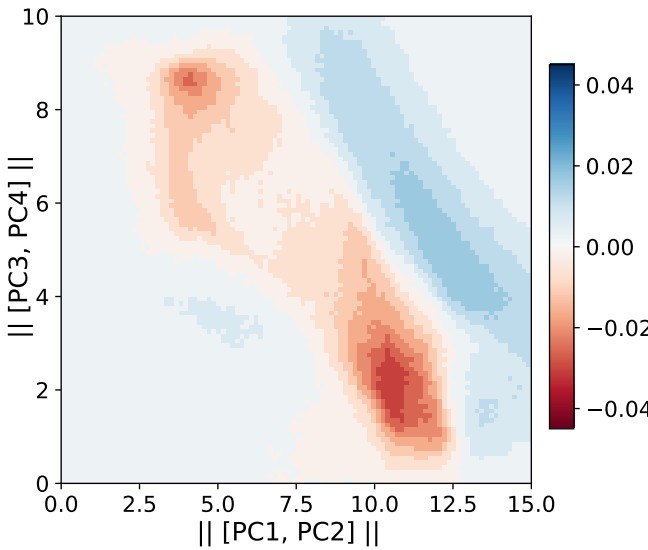

**Figure 8.** Difference between the $F = 21.5$ and $F = 28$ densities in principal component space. The principal components are those from the $F = 20$ truth series. 100 bins are used for this visualisation.

In all the below experiments the polynomial model's trajectories exploded (went to infinite values) so the polynomial is omitted. This is merely due to the specification of a third-order polynomial. It significantly deviates from its target values when $X_{k,t}$ values notably different from those in training (such as $X_{k,t} \geq 19$) are taken as inputs.

The $F = 28$ climate was explored in a similar manner to the $F = 20$ one. First, for the histograms of $X_{k,t}$ (analogous to Figure 3) the L96-RNN and GAN both had small KL divergences (0.02 vs 0.14). Next, the principal component projections were examined (analogous to Figure 4). The same components as determined from the $F = 20$ truth data set are used. In this space, the L96-RNN has a smaller KL-divergence (11) than the GAN (57). Figures for the above two plots are omitted due to it being hard to visually distinguish the models. As before, further comparison can be made by examining the two, 1D density plots in Figure 9 (analogous to Figure 5). Both models perform well (KL divergence in Table 1), but neither put enough density in the right-hand side tails. In terms of temporal correlation, as seen in Figure 6(b) and Figure 7(b), both models perform well, but the L96-RNN generally better tracks the periodicity and the amplitude of the peaks and troughs.

The models were also evaluated in the weather forecasting framework for $F = 28, 32, 35$ and $40$. 750 initial conditions were randomly selected from the truth attractors and an ensemble of 40 forecasts each lasting 2.5 MTU were produced. Figure 10 shows the L96-RNN generally has lower errors than the GAN, with a better matching spread. The GAN continues to be underdispersive.

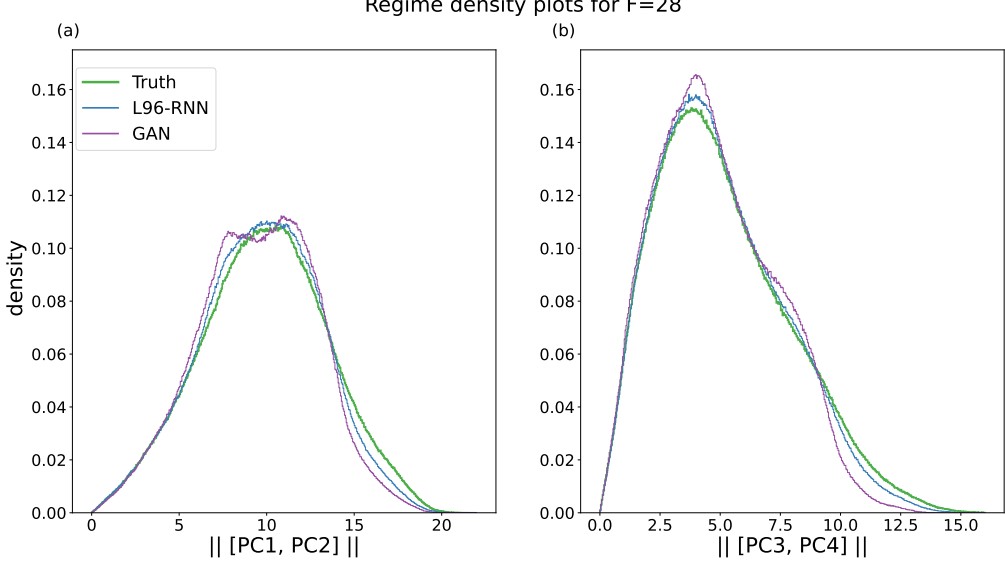

**Figure 9.** Density plots for each regime for $F = 28$. The truth is shown in green and **bold**. (a) Magnitude of $[PC1,PC2]$. 479 bins are used. (b) Magnitude of $[PC3,PC4]$. 444 bins are used. The right-hand side tails are not given sufficient density by the L96-RNN nor the GAN.

### 4.4 Residual Analysis

We can also compare the polynomial and L96-RNN's ability to capture temporal patterns by assessing their residuals. The residuals are the differences between observed data and a model's predictions. It can be considered an offline metric. For the proposed models, the residuals should be uncorrelated and resemble white noise. In Figure 11 we show the autocorrelation of the residuals. Ideally, there would be no significant autocorrelation. Both models do not succeed in this aspect, but the autocorrelation of the L96-RNN's residuals drops off far quicker than that of the polynomial. This is likely due to the L96-

RNN's ability to capture longer temporal trends. The L96-RNN still has a decaying oscillation, suggesting further changes must be made to the model to account for this.

### 4.5 Computational Costs

We wish our models to have a lower simulation cost than the full L96. This is measured by considering the number of floating-point operations per simulated time step ($\Delta t = 0.005$). The L96-RNN (8,682) is notably cheaper than the truth model (88,000)

and the GAN (14,074), so meets the computational cost objective of parameterization work. The polynomial (334) is the cheapest.

In terms of training times, both ML models were trained using a 32GB NVIDIA V100S GPU. The L96-RNN and GAN took 12 minutes and 30 hours respectively.

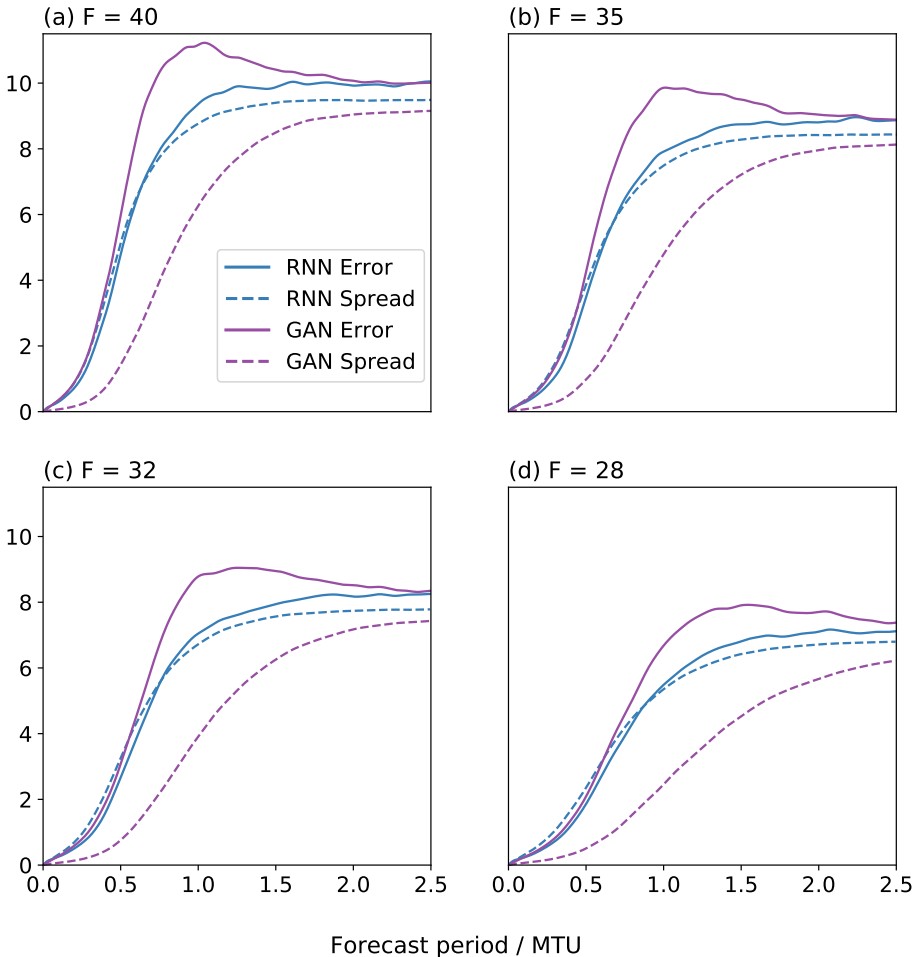

**Figure 10.** Error and spread for weather experiments for varying forcing values. This is done to assess generalisation performance. The L96-RNN's spread is best matched to its error unlike the GAN.

## 5 Evaluation and Diagnostics Using Likelihood

### 5.1 Hold-out Likelihood

We also evaluate using the likelihood (explained in Section 3.2) of hold-out data. This is a standard approach in the ML literature. Evaluation is done on hold-out sets to ensure that overly complex models which just overfit the training data are not selected.

Here, hold-out sets for $F = 20$ and $F = 28$ were created by taking a 10,000 MTU subset from the 50,000 MTU truth sets created in Section 4. The hold-out log-likelihoods are shown in Table 2. The likelihood for the polynomial model and its

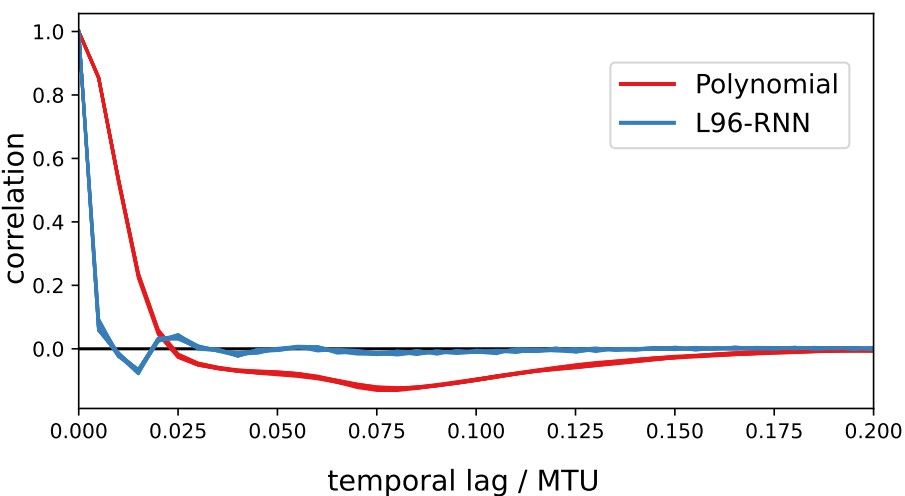

**Figure 11.** Autocorrelation plot of the residuals for the polynomial and L96-RNN. Ideally, these would show no correlations (and match the black line). The L96-RNN's residual correlations decay quicker than the polynomial's.

derivation is in Appendix A. We also present an approach to approximate the GAN likelihood. This is despite its full form being intractable (involving integrals which cannot be efficiently approximated using Monte Carlo sampling) and so it not being typically used to evaluate GANs. This is also detailed in Appendix A. The L96-RNN has a worse likelihood on $F = 28$ than the polynomial, despite the polynomial simulations exploding unlike the L96-RNN's. This is discussed below.

## 5.2 What is the Use of Likelihood?

### 5.2.1 For Evaluation

The other metrics used above only capture snapshots of model performance. Likelihood is a composite measure which assesses a model's full joint distribution (Casella and Berger, 2002). For example, in the univariate case, the mean-squared-error and variance of a model tell two separate things about its performance, with implicit assumptions being made about variables being normally distributed (as is the case whenever mean-squared-error is used). The likelihood captures both of these, and without enforcing such assumptions.

Likelihood also captures information about more complex, joint distributions, saving the need for custom metrics to be invented to assess specific features. To illustrate this, consider we wish to assess a model's temporal associations (not just the linear temporal correlations). The likelihood already contains this information. Although custom metrics could be invented to assess this, the likelihood is an off-the-shelf metric which is already available. We suggest it is wasteful not to use it.

Being a composite metric brings challenges though. Poor performance in certain aspects may be overshadowed by good performance elsewhere. This can result in cases where increased hold-out likelihoods do not correspond to better sample

**Table 2.** *Log-likelihood on hold-out data for different forcing values.*

| Forcing | Polynomial | L96-RNN | GAN |
|---------|-----------|---------|-----|
| 20 | 4.98 | **6.53** | $\approx -1 \times 10^8 \pm 0 \times 10^8$ |
| 28 | **4.07** | 2.69 | $\approx -5 \times 10^8 \pm 0 \times 10^8$ |

quality, as noted in the ML literature (Goodfellow et al., 2014; Grover et al., 2018; Theis et al., 2015; Zhao et al., 2020) and seen in our results: the L96-RNN has a worse $F = 28$ hold-out likelihood than the polynomial (Table 2), yet the polynomial model explodes unlike the L96-RNN. In this case, the phenomenon of 'explosion' is not significantly penalising the likelihood. This links to how likelihood is evaluated on trajectories of the true system, so is an 'offline' metric. Just as with the offline metric of mean-squared-error, you would expect better 'online' performance (the performance of generated simulations) as the likelihood improves, but there is no guarantee of a monotonic relationship between the two. Likelihood is therefore a complement, not replacement, to other metrics which are important to the end-user.

### 5.2.2 For Diagnostics

Likelihood's composite nature makes it a helpful diagnostic tool. Just like KL-divergence, the further away a model is — in any manner — from the data in the hold-out set, the worse the likelihood will be. If a model has a poor likelihood despite performing well on a range of standard metrics, this suggests there are still deficiencies in the model which need investigating. For example, with the L96-RNN at $F = 28$, the poor average likelihood (Table 2) is caused by a tiny number of segments of the $X_{k,t}$ sequences being extremely poorly modelled (and therefore having extremely poor likelihoods assigned). On inspection, the issue is due to the choice of a fixed $\sigma$ in equation (11) — this is appropriate for most of the time series, but for parts which are difficult to model it is too small, preventing the model from expressing sufficient uncertainty. This could be rectified by allowing $\sigma$ to vary, and we suggest future work explore this.

## 6 Discussion and Conclusion

We present an approach to replace red noise with a more flexible stochastic machine-learnt model for temporal patterns. Even though we used ML to model the deterministic part $g_\theta(X_{k,t})$ in equation (10), the real benefit came from using ML to model the hidden variables. For example, on setting $g_\theta(X_{k,t})$ in our model equal to $aX_{k,t}^3 + bX_{k,t}^2 + cX_{k,t} + d$ from the polynomial baseline, our model performance hardly suffered (and the cost halved) for $F = 20$. This finding is supportive of physics-based ML approaches where conventional parameterizations are augmented with a stochastic term learnt using ML.

Using physical knowledge to structure ML models can help with learning. The L96-RNN includes 'physically-relevant' features for the L96, particularly advection, in equation (10). This gave better results than when we modelled the system without them (e.g. when we used an RNN to do the full updating step of $\mathbf{X}_t$ given $\mathbf{X}_{t-1}, ..., \mathbf{X}_0$). Although in theory a NN can *learn* to create helpful features (such as advection), giving these useful features can make learning easier. The NN does not need to learn the known, useful physical relationships and instead can focus on learning other helpful ones.

There are more sophisticated models which can be used for the L96 system. We noticed that in some cases the L96-RNN struggled even to overfit the training data, regardless of the L96-RNN complexity. This could be due to difficulties in learning the evolution of the hidden variables. Creating architectures that permit better hidden variables to be learnt (ones which model long-term correlations better) would give better models. Despite our use of GRU cells, which along with the LSTM were great achievements in sequence modelling, we still faced issues with the modelling of long-term trends as evidenced in Figure 11. Attention-based models such as the Transformer (Vaswani et al., 2017) may improve on this. Separately, certain models (mainly those which did not leverage physical structure) resulted in unstable simulations. Using hierarchical models which learn to model trends at different temporal resolutions (Chung et al., 2016; Liu et al., 2020) may provide improvement. Our particular model had specific limitations in how the hidden state, $R_t$, was not a function of $X_{t-1}$, unlike the GAN, and in how $\sigma$ was not a function of other variables. We did this to make a clearer comparison between hidden variables modelled with AR1 processes versus those modelled with RNNs. Nevertheless, it is simple to include $X_{t-1}$ into the model for $R_t$, and to make $\sigma$ a neural network function of other variables (including the forcing) if desired. We believe the latter change would be particularly useful in better capturing uncertainty.

We trained the L96-RNN using the likelihood of the sequence $(\mathbf{x}_0, \mathbf{x}_1, ..., \mathbf{x}_n)$ and showed how for a RNN-based model, there is an explicit form of the likelihood which makes likelihood calculations tractable. In the L96, these $\mathbf{X}_t$ sequences are K-dimensional and correlated. Their likelihood can be written in terms of the likelihood of the hidden variables $(\mathbf{r}_1, \mathbf{r}_2, ..., \mathbf{r}_n)$, and as shown in equation (13), training to maximize the likelihood of a sequence of hidden variables is equivalent to maximizing the likelihood of a $\mathbf{X}_t$ sequence. However, in the general case of a general circulation model, it will often not be possible to analytically derive the relationship between the likelihood of a $\mathbf{X}_t$ sequence and that for a sequence of hidden variables, as explained in Section A4.1. Therefore, maximizing the likelihood of other variables (the approach typically taken), such as the hidden variables or sub-grid terms, may not correspond to maximizing the likelihood of $\mathbf{X}_t$. This is unideal as our actual goal is to create a model which generates realistic sequences of $\mathbf{X}_t$. And this may be a reason why online evaluations of such models often show error accumulation leading to simulation blow-ups (Brenowitz and Bretherton, 2018; Rasp, 2020). In future work, we plan to investigate how to better train parameterization models for cases where you cannot easily maximize a likelihood which is proportional to the likelihood of $\mathbf{X}_t$.

Learning from all the high-resolution data whilst still being computationally efficient at simulation time is an interesting idea. Whilst we cannot keep all the high-resolution variables in our schemes (as otherwise we might as well run the high-resolution model outright — which is not possible as it is too slow for the desired timescales and is a reason why this field of work exists), only keeping the average (as in existing work which coarse-grains data) means much data is thrown away. Our preliminary investigations used the graphical model in Figure 12. We designed this so that in training, $\mathbf{l}_t$ is learnt such that it contains useful information about the high-resolution data, $\mathbf{Y}_t$, whereas during generation, $\mathbf{Y}_t$ is not required to be simulated. For the L96, this showed no improvement over our L96-RNN though. Parthipan and Wischik (2022) show that such an approach may be useful as a regulariser when training data is limited.

Further work with GANs could result in better models. There may be aspects of realism which we either may not be aware of or may not be able to quantify, so are difficult to include explicitly in our probability models. The GAN discriminator could

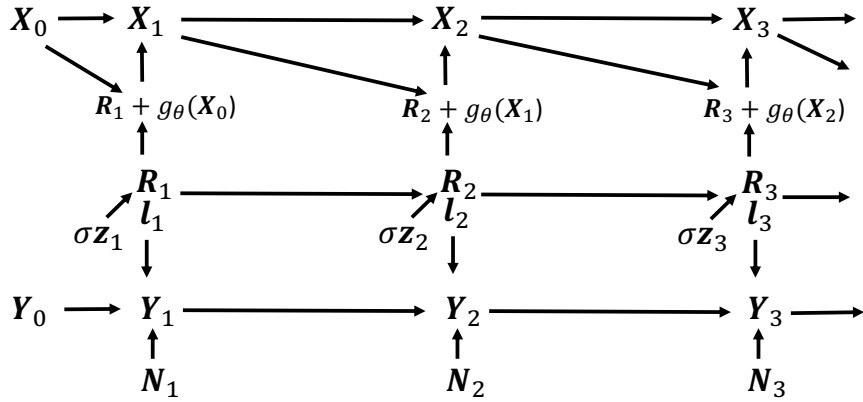

**Figure 12.** Model which allows learning from high-resolution data without requiring it at simulation time, where $N_t \sim \mathcal{N}(0, I)$. The model is trained to generate $Y$, and so learns from this data, but at simulation time, $Y$ is not required to generate $X$.

learn what features constitute 'realism' and so the generator may learn to create sequences which contain these. However, we found that using the advesarial approaches from GANs and Wasserstein-GANs (Arjovsky et al., 2017; Gulrajani et al., 2017) to train the L96-RNN (instead of by likelihood) did not show benefit. This may be due to the difficulty of training GANs on longer sequences. They have been used to train RNNs on medical timeseries (Esteban et al., 2017) and to model music (Mogren, 2016) so applying this approach to climatic sequences may be something for future work to reconcile. The challenges of using GANs are seen by how we have not been able to reproduce the results shown in Gagne et al. (2020) despite the same set-up. This points to the instability of GAN training as noted by them too. Mode collapse is a common issue encountered when training GANs, where the generator fails to produce samples that explore certain modes of the distribution, and could explain why density is only present for Regime A in Figure 4 and not for Regime B.

### 6.1 Conclusion

We have shown how to build on the benefits of red noise models (such as AR1 ones) in parameterizations by using a probabilistic ML approach based on a Recurrent Neural Network (RNN). This can be seen as a natural generalization of the classical autoregressive approaches. This is done in the idealized case of the two-level Lorenz 96 system, where the unresolved variables must be parameterized. Our L96-RNN outperforms the red-noise baselines (a polynomial model and a GAN) in a weather forecasting framework, and has lower KL-divergence scores for the probability density functions arising from long-range simulations. The L96-RNN generalizes the best to new forcing scenarios. These strong empirical results, along with a less correlated residual pattern, supports the benefit of the L96-RNN being due to it better capturing temporal patterns.

This approach now needs testing on more complex systems — both to specifically improve on AR1 processes, and to learn new, more flexible models of $\mathbf{X}_t$. An interesting area for further work is to examine what information is tracked in the hidden variables of the L96-RNN, and how this relates to the timescales of the required memory in the modelled system. The field is

415 ripe for other probabilistic ML tools to be used and we suspect that further customisation of these will lead to many improved parameterization models.

Likelihood can often be fairly easily calculated, and where this be the case, we propose that the community also evaluate the hold-out likelihood for any devised probabilistic model (ML or otherwise). It is a useful debugging tool, assessing the full joint distribution of a model. It would also provide a consistent evaluation metric across the literature. Given the challenges relating

to sample quality, likelihood should complement, not replace, existing metrics.

Finally, we have used demanding tests to show that ML models can generalise to unseen scenarios. We cannot hope for ML models to generalise to all settings. We do not expect this from our physics-based models either. But ML models *can* generalise outside of their training realm. And it is by using challenging hold-out tests that we can assess their ability to do so, and if they fail, begin the diagnosis of why.

*Code and data availability.* All code used in this study (including the code required to create the data) is publicly available at https://doi.org/10.5281/zenodo.7118667.

## Appendix A: Likelihood derivations

In all cases, the log-likelihood of the sequence of $\mathbf{X}_t$ is given by

$$\log \Pr(\mathbf{x}_1, ..., \mathbf{x}_n; \mathbf{x}_0) = \log \Pr(\mathbf{x}_1; \mathbf{x}_0) + \log \Pr(\mathbf{x}_2 | \mathbf{x}_1; \mathbf{x}_0) + \log \Pr(\mathbf{x}_3 | \mathbf{x}_2, \mathbf{x}_1; \mathbf{x}_0)$$

$$+ ... + \log \Pr(\mathbf{x}_n | \mathbf{x}_{n-1}, ..., \mathbf{x}_1; \mathbf{x}_0) \tag{A1}$$

which follows from the laws of probability.

### A1  L96-RNN

Now from (10), we can write

$$\mathbf{X}_t | \mathbf{x}_{t-1}, ..., \mathbf{x}_1; \mathbf{x}_0 = f(\mathbf{x}_{t-1}) - \Delta t (\mathbf{R}_t | \mathbf{x}_{t-1}, ..., \mathbf{x}_1; \mathbf{x}_0) \tag{A2}$$

where for the L96-RNN

$$f(\mathbf{x}_t) = \mathbf{x}_t + \omega(\mathbf{x}_t) - \Delta t \, g_\theta(\mathbf{x}_t) \tag{A3}$$

and $\mathbf{X}_t$ denotes the random variable and $\mathbf{x}_t$ denotes the value which that random variable takes. Given this relationship between $\mathbf{X}_t$ and $\mathbf{R}_t$, a change of variables (further details are in Appendix A4) for the likelihood gives

$$\log \Pr(\mathbf{x}_t | \mathbf{x}_{t-1}, ..., \mathbf{x}_1; \mathbf{x}_0) = \log \Pr(\mathbf{r}_t | \mathbf{r}_{t-1}, ..., \mathbf{r}_1) - K \log(\Delta t) \tag{A4}$$

(A1) is therefore

$$= \log \Pr(\mathbf{r}_1) + \log \Pr(\mathbf{r}_2|\mathbf{r}_1) + \log \Pr(\mathbf{r}_3|\mathbf{r}_2, \mathbf{r}_1)$$
$$+ ... + \log \Pr(\mathbf{r}_n|\mathbf{r}_{n-1:1}) - Kn \log(\Delta t) \tag{A5}$$

$$= \log \Pr(\mathbf{r}_1) + \sum_{t=2}^{n} \log \Pr(\mathbf{r}_t|\mathbf{l}_t) - Kn \log(\Delta t) \tag{A6}$$

$$= \sum_{k=1}^{K} \left( \log \Pr(r_{k,1}) + \sum_{t=2}^{n} \log \Pr(r_{k,t}|l_{k,t}) \right) - Kn \log(\Delta t) \tag{A7}$$

where (A6)–(A7) follow from the independencies in the graphical model (Figure 1).

## A2  Polynomial Model

For the polynomial model, the log-likelihood of the $\mathbf{X}_t$ sequence is

$$\log \Pr(\mathbf{x}_1, ..., \mathbf{x}_n; \mathbf{x}_0) = \sum_{k=1}^{K} \left( \log \Pr(h_{k,1}) + \sum_{t=1}^{n-1} \log \Pr(h_{k,t+1}|h_{k,t}) \right) - Kn \log(\Delta t) \tag{A8}$$

and the derivation follows below.

The graphical model is shown in Figure A1a. From (6), we can write

$$\mathbf{X}_t|\mathbf{x}_{t-1}, ..., \mathbf{x}_1; \mathbf{x}_0 = f(\mathbf{x}_{t-1}) - \Delta t(\mathbf{H}_t|\mathbf{x}_{t-1}, ..., \mathbf{x}_1; \mathbf{x}_0) \tag{A9}$$

and given this relationship between $\mathbf{X}_t$ and $\mathbf{h}_t$, a change of variables for the likelihood gives

$$\log \Pr(\mathbf{x}_t|\mathbf{x}_{t-1}, ..., \mathbf{x}_1; \mathbf{x}_0) = \log \Pr(\mathbf{h}_t|\mathbf{h}_{t-1}, ..., \mathbf{h}_1) - K \log(\Delta t) \tag{A10}$$

(A1) is therefore

$$= \log \Pr(\mathbf{h}_1) + \log \Pr(\mathbf{h}_2|\mathbf{h}_1) + \log \Pr(\mathbf{h}_3|\mathbf{h}_2, \mathbf{h}_1)$$
$$+ ... + \log \Pr(\mathbf{h}_n|\mathbf{h}_{n-1:1}) - Kn \log(\Delta t) \tag{A11}$$

$$= \log \Pr(\mathbf{h}_1) + \sum_{t=1}^{n-1} \log \Pr(\mathbf{h}_{t+1}|\mathbf{h}_t) - Kn \log(\Delta t) \tag{A12}$$

$$= \sum_{k=1}^{K} \left( \log \Pr(h_{k,1}) + \sum_{t=1}^{n-1} \log \Pr(h_{k,t+1}|h_{k,t}) \right) - Kn \log(\Delta t) \tag{A13}$$

where (A12)–(A13) follow from the independencies described in (5).

## A3  GAN

We approximate the GAN's likelihood by calculating the likelihood of a model which functions and gives results almost
455 identical to the GAN (one with a small amount of white noise added) using importance sampling and the reparameterization

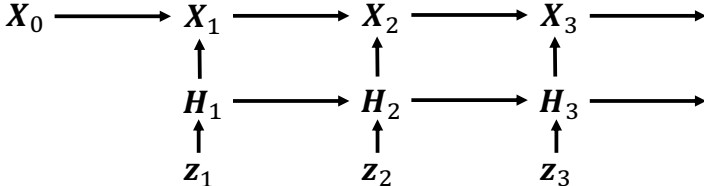

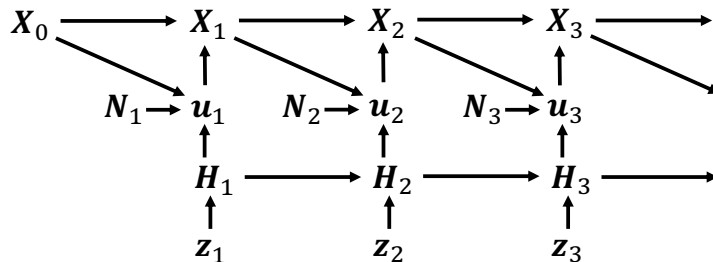

**Figure A1.** Graphical model for (a) Polynomial and (b) GAN with added white noise.

method as in the Variational Autoencoder Kingma and Welling (2013). Here, (9) is altered to

$$X_{k,t+1} = X_{k,t} + \omega_k(\mathbf{X}_t) - \Delta t \, u(X_{k,t}, H_{k,t+1}) \tag{A14}$$

where $u(X_{k,t}, H_{k,t+1}) = U(X_{k,t}, H_{k,t+1}) + N_{k,t+1}$ and $N_{k,t} \sim \mathcal{N}(0, 0.001^2)$. The log-likelihood is

$$\log \Pr(\mathbf{x}_1, ..., \mathbf{x}_n; \mathbf{x}_0) = \log \mathbb{E}_{\mathbf{h} \sim \Pr_{\tilde{\mathbf{H}}.}} \frac{\Pr(\mathbf{u}. | \mathbf{h}., \mathbf{x}_0) \Pr_{\mathbf{H}.}(\mathbf{h}.)}{\Pr_{\tilde{\mathbf{H}}.}(\mathbf{h}.)} - Kn \log(\Delta t)$$

where the expectation can now be approximated by a large sum if there is a suitable importance sampler $\Pr_{\tilde{\mathbf{H}}.}(\mathbf{h}.)$. 50 samples were used for the importance sampling, and this was repeated 50 times to give 95% confidence intervals. We note that given the finite number of samples taken it is of course possible that the GAN likelihood is larger, but a well-trained importance sampler should minimise the chance of this. The derivation and form of the importance sampler is detailed below.

The graphical model of the GAN with a small amount of white noise added is shown in Figure A1b. $\mathbf{X}_t$ is related to $\mathbf{u}_t$ as in (A14) so

$$(\mathbf{X}_t | \mathbf{X}_{t-1} = \mathbf{x}_{t-1}, ..., \mathbf{X}_1 = \mathbf{x}_1; \mathbf{X}_0 = \mathbf{x}_0) = f(\mathbf{x}_{t-1}) - \Delta t(\mathbf{u}_t | \mathbf{X}_{t-1} = \mathbf{x}_{t-1}, ..., \mathbf{X}_1 = \mathbf{x}_1; \mathbf{X}_0 = \mathbf{x}_0) \tag{A15}$$

and a change of variables for likelihood gives

$$\log \Pr(\mathbf{x}_t | \mathbf{x}_{t-1}, ..., \mathbf{x}_1; \mathbf{x}_0) = \log \Pr(\mathbf{u}_t | \mathbf{u}_{t-1}, ..., \mathbf{u}_1; \mathbf{x}_0) - K \log(\Delta t) \tag{A16}$$

(A1) is therefore

$$= \log \Pr(\mathbf{u}_1|\mathbf{x}_0) + \log \Pr(\mathbf{u}_2|\mathbf{u}_1, \mathbf{x}_0) + \log \Pr(\mathbf{u}_3|\mathbf{u}_2, \mathbf{u}_1, \mathbf{x}_0)$$

$$+ ... + \log \Pr(\mathbf{u}_n|\mathbf{u}_{n-1:0}, \mathbf{x}_0) - Kn \log(\Delta t)$$

$$= \log \Pr(\mathbf{u}.|\mathbf{x}_0) - Kn \log(\Delta t)$$

and just decomposing the first term below:

$$\log \Pr(\mathbf{u}.|\mathbf{x}_0) = \log \mathbb{E}_{\mathbf{h} \sim \Pr_{\mathbf{H}}}. \Pr(\mathbf{u}.|\mathbf{h}., \mathbf{x}_0)$$

$$= \log \mathbb{E}_{\mathbf{h} \sim \Pr_{\tilde{\mathbf{H}}}.} \frac{\Pr(\mathbf{u}.|\mathbf{h}., \mathbf{x}_0) \Pr_{\mathbf{H}.}(\mathbf{h}.)}{\Pr_{\tilde{\mathbf{H}}.}(\mathbf{h}.)} \tag{A17}$$

where (A17) can be approximated with a sufficiently large sum given a good enough encoder $\Pr_{\tilde{\mathbf{H}}.}(\mathbf{h}.)$.

For training purposes, the lower-bound is used:

$$\geq \mathbb{E}_{\mathbf{h} \sim \Pr_{\tilde{\mathbf{H}}.}} \log \frac{\Pr(\mathbf{u}.|\mathbf{h}., \mathbf{x}_0) \Pr_{\mathbf{H}.}(\mathbf{h}.)}{\Pr_{\tilde{\mathbf{H}}.}(\mathbf{h}.)} \tag{A18}$$

The terms in the numerator decompose as follows, using the independencies from the graphical model and associated equa-
tions:

$$\log \Pr(\mathbf{u}.|\mathbf{h}., \mathbf{x}_0) = \sum_{t=1}^{n} \log \Pr(\mathbf{u}_t|\mathbf{h}_t, \mathbf{x}_{t-1})$$

$$= \sum_{t=1}^{n} \sum_{k=1}^{K} \log \Pr(u_{k,t}|h_{k,t}, x_{k,t-1})$$

and

$$\log \Pr_{\mathbf{H}.}(\mathbf{h}.) = \log \Pr_{\mathbf{H}}(\mathbf{h}_1) + \sum_{t=2}^{n} \log \Pr_{\mathbf{H}}(\mathbf{h}_t|\mathbf{h}_{t-1})$$

$$= \sum_{k=1}^{K} \left( \log \Pr_H(h_{k,1}) + \sum_{t=2}^{n} \log \Pr_H(h_{k,t}|h_{k,t-1}) \right)$$

The perfect importance sampling distribution would be proportional to $\Pr(\mathbf{h}.|\mathbf{u}., \mathbf{x}_0)$. Therefore we want $\Pr_{\tilde{\mathbf{H}}.}(\mathbf{h}.)$ to allow
for the same dependencies. We choose it by carrying out the following steps:

$$\log \Pr_{\tilde{\mathbf{H}}.}(\mathbf{h}.) \propto \log \Pr(\mathbf{h}.|\mathbf{u}., \mathbf{x}_0)$$

$$= \log \Pr_{\mathbf{H}_1}(\mathbf{h}_1|\mathbf{u}., \mathbf{x}_0) + \sum_{t=2}^{n} \log \Pr_{\mathbf{H}_t}(\mathbf{h}_t|\mathbf{h}_{t-1}, \mathbf{u}., \mathbf{x}_0) \tag{A19}$$

where (A19) follows from using the laws of probability to decompose the term above, followed by applications of the inde-
pendencies from the graphical model. Note that although the process is Markov it is not necessarily time-homogeneous. To

deal with this, an RNN is used to model $\Pr_{\mathbf{H}_t}(\mathbf{h}_t | \mathbf{h}_{t-1}, \mathbf{u}., \mathbf{x}_0)$ as $\Pr_{\mathbf{H}}(\mathbf{h}_t | \mathbf{h}_{t-1}, \mathbf{u}., \mathbf{x}_0, \mathbf{s}_t)$ where the state $\mathbf{s}_t$ can in principle account for the non-homogeneous updating procedure of $\mathbf{h}_t$. The log-likelihood is therefore

$$= \sum_{k=1}^{K} \left( \log \Pr_{H_1}(h_{k,1}|u_{k.}, x_{k,0}) + \sum_{t=2}^{n} \log \Pr_{H_t}(h_{k,t}|h_{k,t-1}, u_{k.}, x_{k.}, s_{k,t}) \right) \tag{A20}$$

where the spatial independencies introduced in (A20) follow from the graphical model and $u_{k,.} = u_{k,1:n}$ and $x_{k,.} = x_{k,0:n-1}$. The only simplifying assumption used here is that the same update model for $h_{k,t}$ is used for each component $k$. Finally, the sequence of $(u_{k.}, x_{k.})$ is summarised using a bidirectional RNN to give $w_{k,t}$. Therefore, the final form of the log-likelihood of our importance sampler is

$$= \sum_{k=1}^{K} \left( \log \Pr_{H_1}(h_{k,1}|w_{k,1}, x_{k,0}) + \sum_{t=2}^{n} \log \Pr_{H_t}(h_{k,t}|h_{k,t-1}, w_{k,t}, x_{k,t-1}, s_{k,t}) \right) \tag{A21}$$

The training of the importance sampler is done by maximizing (A18), with the learnable parameters being those of the model used to learn $h_{k,1}$, the RNN used to model the update of $h_{k,t}$, and the bidirectional RNN.

## A4 Change of variables for probability density functions

For random variables, $X \in \mathbb{R}^1$ and $Y \in \mathbb{R}^1$, where $X = \eta(Y)$, and $\eta : \mathbb{R}^1 \to \mathbb{R}^1$, the probability density function for $X$, $\Pr_X(x)$, is

$$\Pr_X(x) = \Pr_Y\left(\eta^{-1}(x)\right) \left| \frac{d}{dx}\left(\eta^{-1}(x)\right) \right| \tag{A22}$$

Now we will show how we arrive at equation (A4) for the RNN. A similar approach can be taken for the other two models. We can rewrite the equation (a single component of equation (A2))

$$X_{k,t}|\mathbf{x}_{k,t-1}, ..., \mathbf{x}_{k,1}; \mathbf{x}_{k,0} = f_k(\mathbf{x}_{t-1}) - \Delta t (R_{k,t}|\mathbf{x}_{k,t-1}, ..., \mathbf{x}_{k,1}; \mathbf{x}_{k,0}) \tag{A23}$$

as

$$X_{k,t}|\mathbf{x}_{k,t-1}, ..., \mathbf{x}_{k,1}; \mathbf{x}_{k,0} = \eta\left(R_{k,t}|\mathbf{x}_{k,t-1}, ..., \mathbf{x}_{k,1}; \mathbf{x}_{k,0}\right) \tag{A24}$$

where

$$\eta\left(R_{k,t}|\mathbf{x}_{k,t-1}, ..., \mathbf{x}_{k,1}; \mathbf{x}_{k,0}\right) = f_k(\mathbf{x}_{t-1}) - \Delta t (R_{k,t}|\mathbf{x}_{k,t-1}, ..., \mathbf{x}_{k,1}; \mathbf{x}_{k,0}) \tag{A25}$$

From this we see

$$\eta^{-1}\left(X_{k,t}|\mathbf{x}_{k,t-1}, ..., \mathbf{x}_{k,1}; \mathbf{x}_{k,0}\right) = \frac{-1}{\Delta t}\left(X_{k,t}|\mathbf{x}_{k,t-1}, ..., \mathbf{x}_{k,1}; \mathbf{x}_{k,0} - f_k(\mathbf{x}_{t-1})\right) \tag{A26}$$

and therefore

$$\frac{d}{dx_{k,t}}\eta^{-1}\left(X_{k,t} = x_{k,t}|\mathbf{x}_{k,t-1}, ..., \mathbf{x}_{k,1}; \mathbf{x}_{k,0}\right) = \frac{-1}{\Delta t} \tag{A27}$$

We can thus write

$$\Pr_{X_{k,t}}(x_{k,t}|\mathbf{x}_{k,t-1},...,\mathbf{x}_{k,1};\mathbf{x}_{k,0}) = \Pr_{R_{k,t}}(r_{k,t}|\mathbf{x}_{k,t-1},...,\mathbf{x}_{k,1};\mathbf{x}_{k,0})\left|\frac{-1}{\Delta t}\right| \tag{A28}$$

$$= \Pr_{R_{k,t}}(r_{k,t}|\mathbf{r}_{k,t-1},...,\mathbf{r}_{k,1})\frac{1}{\Delta t} \tag{A29}$$

And from the conditional independencies in equations (10)-(12),

$$\Pr_{\mathbf{X}_t}(\mathbf{x}_t|\mathbf{x}_{t-1},...,\mathbf{x}_1;\mathbf{x}_0) = \prod_{k=1}^{K}\Pr_{X_{k,t}}(x_{k,t}|\mathbf{x}_{k,t-1},...,\mathbf{x}_{k,1};\mathbf{x}_{k,0}) \tag{A30}$$

$$= \prod_{k=1}^{K}\Pr_{R_{k,t}}(r_{k,t}|\mathbf{r}_{k,t-1},...,\mathbf{r}_{k,1})\frac{1}{\Delta t} \tag{A31}$$

$$= \Pr_{R_t}(\mathbf{r}_t|\mathbf{r}_{t-1},...,\mathbf{r}_1)\left(\frac{1}{\Delta t}\right)^{K} \tag{A32}$$

and on taking logs and dropping the subscripts we get

$$\log\Pr(\mathbf{x}_t|\mathbf{x}_{t-1},...,\mathbf{x}_1;\mathbf{x}_0) = \log\Pr(\mathbf{r}_t|\mathbf{r}_{t-1},...,\mathbf{r}_1) - K\log(\Delta t) \tag{A33}$$

which is equation (A4).

### A4.1 Application to systems beyond the L96

It will sometimes not be possible to apply the changes of variables approach to more complex systems, for a few reasons. First, it requires the function $\eta$ in $X = \eta(Y)$ to be either a monotonically increasing or monotonically decreasing function (as the inverse must exist). This may not be the case in more sophisticated climate models. Second, the derivative of the inverse of $\eta$ needs to be computable, which may not be feasible to do if the inverse is too challenging to calculate (for example, it may not be possible to run individual components of climate model functions in order to calculate $\eta^{-1}$).

### Appendix B: Further results

Further results for KL-divergences are provided in order to see how results vary with differing bin sizes for the histograms.

Table B1 is for the distributions in Figure 3, and Table B2 is for those in Figure 4. Even for a range of bin sizes, the L96-RNN has the lowest KL-divergences.

*Author contributions.* All authors supported the design of the study. RP created the models and conducted research. RP prepared the manuscript with contributions from all co-authors.

*Competing interests.* The authors declare that they have no conflict of interest.

**Table B1.** *KL-divergence (goodness-of-fit) between 1) the truth and 2) the parameterized models, for the distributions shown in Figure 3. The smaller the KL-divergence, the better the match between the true and modelled distributions. The best model in each case is shown in* **bold**.

| Number of bins | Polynomial | L96-RNN | GAN |
|:---:|:---:|:---:|:---:|
| 600 | 0.09 | **0.03** | 0.60 |
| 700 | 0.11 | **0.04** | 0.70 |
| 800 | 0.12 | **0.04** | 0.80 |
| 827 | 0.13 | **0.04** | 0.83 |
| 900 | 0.14 | **0.05** | 0.90 |
| 1000 | 0.16 | **0.05** | 1.01 |

**Table B2.** *KL-divergence (goodness-of-fit) between 1) the truth and 2) the parameterized models, for the distributions shown in Figure 4. The smaller the KL-divergence, the better the match between the true and modelled distributions. The best model in each case is shown in* **bold**.

| Number of bins | Polynomial | L96-RNN | GAN |
|:---:|:---:|:---:|:---:|
| 250 | 29 | **14** | 345 |
| 350 | 64 | **32** | 735 |
| 450 | 123 | **65** | 1310 |
| 550 | 218 | **122** | 2111 |
| 650 | 366 | **214** | 3178 |

*Acknowledgements.* We would like to thank Pavel Perezhogin and our other anonymous reviewer for their feedback which has improved the manuscript. R.P. was funded by the Engineering and Physical Sciences Research Council [grant number EP/S022961/1]. H.M.C. was supported by the Natural Environment Research Council [grant number NE/P018238/1]. J.S.H. was supported by the British Antarctic Survey, Natural Environment Research Council (NERC) National Capability funding. D.J.W. was funded by the University of Cambridge.

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
