# Peer review of "Using Probabilistic Machine Learning to Better Model Temporal Patterns in Parameterizations: a case study with the Lorenz 96 model."

_EGUsphere, 2022_

## Referee Comment (RC1)

**Using Probabilistic Machine Learning to Better Model Temporal Patterns in Parameterizations: a case study with the Lorenz 96 model.**

October 2022

The manuscript under consideration implements Recurrent Neural Networks for stochastic parameterization of the Lorenz 96 Model. The authors compare the performance of their model in emulation of the Lorenz 96 to a GAN and a baseline polynomial. Their goal is to capture temporal patterns of this idealized atmosphere better and generalize to out-of-sample atmospheric states. They see this work as building on the more naive addition of red noise to parameterizations for a probabilistic framework. As evidence of this, they present approximated Kull-Back Liebler Divergences and likelihood scores from the various models examined.

There are a number of aspects to this manuscript that make it appealing. It uses modern machine learning techniques (LSTM) to try and address a problem (stochasticity) that previous implementations of machine learning in this space have struggled with. However, this manuscript has significant shortcomings. The most significant of these are the evaluation metrics and baselines the authors use to attempt to prove the significance of their findings. In their present form, it is not clear if we can gain any insight from them. Furthermore, crucial details about model training are missing. There are general issues with readability including incomplete figures, undefined terms, and poorly defined concepts that would make this article difficult for a general audience. For these reasons, I recommend Major Revisions prior to consideration for publication in the Journal of European Geosciences Union (EGU). Below, I give recommendations to address the manuscript's shortcomings

**Contents**

**1  Major Issues**

**1.1  Missing context and clarity in manuscript setup**

The authors do a nice job of framing the issue that small-scale and unresolved processes are the greatest drivers of uncertainty in the cloud-climate feedback. However, given its importance to the premise of the manuscript, the authors need to give an explanation of what exactly "red noise" is rather than just saying it is important to parameterization. Likewise for "white noise" mentioned but not explained (and possibly blue noise as well). This will make the work more approachable to a more general audience. There are other examples of this throughout the manuscript but one more to point out is that, especially given it is the model of choice for the authors, the difference between traditional RNNs and LSTM should be explained more, and how specifically it is helpful in this framework as well as terms like "gated".

**1.2  clarity in Training procedure**

The authors do include some important information regarding the development of their model but more information is needed for reproducibility. The authors should clarify the process of hyper-parameter tuning that lead to their final hyper-parameter choices. What was tried and what wasn't? The authors should also clarify if they did a training/validation/test split or if they just do a training/validation split – and if the latter they should provide some justification for this decision. Additionally, did the authors experiment with different training/validation splits before settling on the final one, and if not a justification should be provided for the absence of this test.

I appreciate the authors details in Section 4.4 about computational costs between different models. But in terms of reproducibility, it would be helpful to include the amount of gpu/cpu/tpu hours required to train and tune these models.

**1.3  Climate valuation Metric lacks justification**

I think the idea of approximating the true continuous distributions through a discrete one is likely appropriate here. But it needs proper justification and explanation (and citations). The authors are deriving the approach through "vector quantization" [2, 4], something they should introduce and explain. But through this vector quantization, the authors are not getting a true KL divergence, but rather estimating the lower bound of the true KL Divergence and their equation in 4.2 (not numbered) should be adjusted to reflect this for accuracy [1].

Because this is an estimation it requires additional justification. The principle (as though be included in this explanation) is that the more "bins" the more accurate the approximation of the lower bound of the KL Divergence. The authors need to justify the number of bins. More specifically they need to include either a graph or table (would recommend including this in the appendix rather than the main text) showing how the approximated KL Divergence changes as bin count is increased and at what bin count the approximated KL Divergence converges. This is the bin count the authors should use. As it stands now we cannot interpret any significance from the results because we don't know if the approximation of the KL Divergence they use is accurate or not.

Additionally, they need to confirm that Figures 3, 5, and 7 are built off of common bins. It would also be appropriate for the authors to acknowledge they are not the first to use this approach, it has been used in both supervised and unsupervised frameworks [1, 3]. Lastly, given that the KL Divergence is non0symmetric, there is an argument that the authors should symmetrize their approximated KL Divergences into Jensen-Shannon Divergences.

**1.4  Missing information**

There are a number of places in the manuscript missing citations, context, or other information or where figures are not complete. I encourage the authors to do a thorough proofread but I will include some of them below.

**2  Technical Edits**

(Line 25-27) [Add citation to paper with an idealized model.]

(Lines 31) [Add a reference to a section/figure that supports this claim.]

(Line 65) [Please be careful about calling any neural networks "deterministic" For most, the stochastic gradient descent and random seedings in initialization can/do add some variance that makes the label "deterministic" inaccurate.]

(Lines 73-73) [Ciation needed for the claim about minmax loss.]

(Lines 104-105) [Is there a physical justification for this in terms of phenomena in the atmosphere? If so, would be helpful to mention that here.]

(Line 105) [Word "good" is opinionated and vague.]

(Line 159) [Language is unscientific.]

(161 and elsewhere) [Would suggest changing the term "truth data" to the more widely used "target data" to avoid confusion.]

(Line 163) [Why is the GAN only trained on $F = 20$? If the authors are trying to claim their model is superior, should it not be trained similarly to the RNN. Otherwise, the scope of the claim needs to be limited to moving beyond the specific paper the GAN was from rather than the approach itself.]

(Figure 2) [Need y axis label and title]

(Figure 3) [Remove ticks on the y-axis of plot b. Add titles.]

(Figures 3, 5, 7) [It is difficult to observe anything quantitative in these plots (as the authors themselves state in line 193). If the authors want to keep them, I would suggest moving to the Appendix.]

(Line 209) [Explicitely define PCx as "Principle Component".]

(Figure 4) [Add a title. Remove x ticks on a,b. Remove y ticks on b, d. This will allow the plots to be a bit bigger. Make the colorbar discrete to match the plots]

(Figure 5) [Remove y ticks from b. Add a title.).]

(Figure 7) [Remove y ticks from b. Add a title.).]

(Figure 5) [Remove y ticks from b, d and x ticks from a.c. Add a title. Set same y axis max on all four plots.).]

(Lines 259-263) [Add citation.]

(Lines 270) [Reference the appropriate table.]

(Lines 277) [Reference the appropriate table.]

(Lines 310-312) [Would suggest including references to diffusion models or VAEs here.]

**References**

[1] J. Duchi. Lecture notes for statistics 311/electrical engineering 377. *Stanford*, 2:23, 2016.

[2] R. Gray. Vector quantization. *IEEE Assp Magazine*, 1(2):4–29, 1984.

[3] G. Mooers, M. Pritchard, T. Beucler, P. Srivastava, H. Mangipudi, L. Peng, P. Gentine, and S. Mandt. Comparing storm resolving models and climates via unsupervised machine learning, 2022.

[4] S. Rabanser, S. Günnemann, and Z. Lipton. Failing loudly: An empirical study of methods for detecting dataset shift. In H. Wallach, H. Larochelle, A. Beygelzimer, F. d'Alché-Buc, E. Fox, and R. Garnett, editors, *Advances in Neural Information Processing Systems*, volume 32. Curran Associates, Inc., 2019.

---

## Referee Comment (RC2)

**Review of "Using Probabilistic Machine Learning to Better Model Temporal Patterns in Parameterizations: a case study with the Lorenz 96 model" by Raghul Parthipan, Hannah M. Christensen, J. Scott Hosking, and Damon J. Wischik**

Pavel Perezhogin

March 4, 2023

The paper is devoted to the development of the stochastic parameterization of the subgrid tendency in an idealized case of the two-level Lorenz 96 system. The ground truth system integrates both levels simultaneously, while the slow variables are considered as a coarse system. The interaction with the fast variables is not directly resolved in the coarse model and needs to be parameterized. There is an uncertainty in the state of the unresolved variables, and thus it is natural to propose a stochastic parameterization of the missing physics. The new stochastic parameterization is intended to improve the red noise model (AR1) of stochastic residuals with the Recurrent Neural Networks (RNN) which can be seen as a natural generalization of the classical AR1 approach. The RNN model is reduces to the AR1 process in the simplest case when neural networks ($b_\theta$ and $s_\theta$) are parameterized with scalar or identity mappings. The major goal of the presented RNN approach is to improve the temporal correlation of the simulated stochastic residuals. The classical RNN model (which is deterministic) is modified with the Gaussian noise in the output layer to make the prediction stochastic. The free parameters of the RNN model are optimized with the maximum likelihood approach. The authors consider the likelihood which is a probability density to observe the trajectory given the probabilistic model with fixed parameters. This likelihood to observe the trajectory is transformed into a function of RNN model variables using the change of variables approach. The likelihood is also considered as an offline metric to evaluate a probabilistic subgrid model. The proposed RNN model is compared to a simple polynomial model and GAN stochastic model. The RNN model outperforms the baselines in the prediction of the spread of the ensemble in the weather forecasting experiment and has a lower KL-divergence metric in online PDFs of the solution and principal components. The generalization to a different forcing shows that in climate online metrics, the RNN and GAN models are similar, and both generalize better than the polynomial parameterization. Generalization in the weather forecasting problem shows superiority of the RNN model with respect to GAN.

The presented approach is a considerable contribution to the research of stochastic parameterizations. My main suggestions are to improve the description of the training procedure and to demonstrate temporal correlation with ACF function. Also, the manuscript can be improved with the discussion of the limitations of the proposed approach. I recommend a **Minor Revision** prior to the publication, and below address the mentioned points in detail.

In case of any questions: pp2681@nyu.edu.

**1    Main issues**

**1.1    RNN model formulation**

The form of the model (Eq. 10-12) may be not familiar to general readers. Below I provide possible details that may be included to the text to improve the readability (up to the author's decision):

- The subgrid parameterization is split into the deterministic part (parameterized by mapping $g_\theta$) and stochastic residual $r$ which is parameterized by RNN.

- The RNN model is given in equations (11-12), and it attempts to predict a sequence of residuals at the next time step $r_{k,t+1}$ given a sequence of $r_{k,t}$.

- The major building block of the RNN is the mapping $s_\theta$, which takes as an input the sequence of residuals $r_{k,t}$ and updates a hidden state $l_{k,t}$ recurrently.

- The output layer of the RNN gives a probabilistic prediction of the residual at the next time step $r_{k,t+1}$ with a Gaussian distribution parameterized by $b_\theta$ and $z$.

- Because sequences $r_{k,t}$ and $r_{k,t+1}$ are related to each other, the stochastic prediction is fed back to the input of the RNN.

Also:

- It would be nice to give a reference to the exactly same form of the RNN model (11-12) if it is possible.

**1.2 Training algorithm**

Line 156: If it is possible, give a reference for the training of RNNs with (similar) likelihood.

The presented description of the training algorithm and loss function (Line 156) are insufficient to reproduce the results. There are missing explanations that:

- The RNN model is trained given the time series of model state $\mathbf{x}_t$ and the true subgrid forcing $U_t$ (or coupling term).

- Be more explicit in "Where $r$ and $l$ are deterministic functions of $\mathbf{x}_0, ..., \mathbf{x}_n$ derived from equations (10)–(12)", particularly:

  – The residual is defined by $r = U_t - g_\theta(x_t)$.
  – $l$ is obtained from equation (12) which is solved given the known history of $r$ and initial condition for $l_0$

- An explicit expression for the loss function is required after changing $\log Pr(r|l)$ by Gaussian probability density with $\log \mathcal{N}(b_\theta(l), \sigma^2) = const - \frac{1}{2\sigma^2}\left(U_t - g_\theta(x_t) - b_\theta(l)\right)^2$. In my belief, the final loss involving most of the trainable mappings and parameters $(g_\theta, b_\theta, \sigma)$, can be helpful to understand the model. For example, it is clear that both deterministic part of the model $(g_\theta)$ and model of stochastic residual $(b_\theta(l))$ are trained to jointly minimize the MSE of subgrid forcing prediction. This joint optimization clearly makes this model superior to the typical approach when first deterministic part is fitted, and then the residual is simulated independently, as in the Polynomial model. This interpretation of the loss may support the statement in Line 285 "For example, ... hardly suffered". I suggest to put the final loss and its interpretation in the Appendix.

**1.3 Temporal correlation**

The full paper is devoted to the temporal correlation, but there is no any plot of the lagged-autocorrelation function (ACF). Provide please ACF for: true subgrid forcing and simulated subgrid forcing (for Polynomial, GAN and RNN), and ACF for true and simulated residuals ($r$ in case of RNN and $h$ in case of Polynomial).

**1.4 Limitations**

Here I suggest topics to be clarified in the Discussion section.

The described approach to train RNN model requires computation of the likelihood of the trajectory $(x_1, x_2, ..., x_n)$, but not the residuals (or subgrid forcing), and it is not common. In the appendix, it is explained that for L96 system both likelihoods can be related to each other (A3):

$$\log Pr(\mathbf{x}_t|\mathbf{x}_{t-1}...\mathbf{x}_1) = \log Pr(\mathbf{r}_t|\mathbf{r}_{t-1}...\mathbf{r}_1) - K\log(\Delta t) \tag{1}$$

It remains unclear if this trick is limited to scalar time series generated by L96. So, the discussion section can be improved with the following topic:

- Does the likelihood of the trajectory remain computationally tractable in the general case of GCM model (many correlated time series).

Also:

- In spite of the joint optimization of deterministic part $g_\theta$ and stochastic residuals $r$, equations for stochastic residuals (11-12) are not informed with the solution $\mathbf{x}_t$. So, the presented model is not conditional (unlike the GAN model). Is it possible to make it conditional?

**2 Technical issues**

- Line 76, "eliminating the need for update functions to be manually specified". Make a reference to $\beta$ or Eq. (3).

- Consider citation of stochastic LSTM model applied to geophysical data [1].

- Line 4: "physically-informed" and Line 289 "physical features", Line 297 "mainly those which did not leverage physical structure" what is meant in these cases?

- Line 9 and in other places "probabilistic metric of hold-out likelihood". Please specify that it is offline metric (computed on the trajectory of the true system).

- Line 41: "invented hidden variables", what does "invented" mean.

- Equation 5: define $\phi$ and $\sigma$ (as lag-one correlation and standard deviation of time series)

- Line 101: in L96 equation for X, check $j, k$ indexes in coupling term.

- Equation (7): change $\lambda(X_t/2)$ to $1/2\lambda(X_t)$

- Line 168: Provide the number of training points in the epoch.

- Line 181: I do not think that $t = t_{init} + \tau$ helps the reader, and most likely wrong.

- Line 182: Why $X_{m,n}(t)$ does not have index "sample", but $X_m^{sample}$ has. They should be the same, I think.

- Line 209: Give a short definition of "in a phase quadrature"

- Line 212: Is $||[PC1, PC2]|| = \sqrt{PC1^2 + PC2^2}$.

- Line 230: Check "The to simulate"

- Line 256: Clarify "despite the polynomial exploding"

- Lines 259-262: Reformulate the whole paragraph.

- Line 289: "the the"

- Figure 9: "Model which allows to learning from high-resolution data" is not clear. The word "learning" should be specified. As I understand, it is proposed to simulate unresolved variables with RNN.

- Line 293: "There are more sophisticated ways to model the system": reformulate.

- Line 310: Probably better to say "GAN loss function to train RNN".

- Check Discussion and Conclusion: the main results of the paper are not stated clearly, i.e. there is no summary.

- Equation (A2): Clear distinction between $\mathbf{x}_t$ and $\mathbf{X}_t$ will help; define the notation for parentheses $(x|y)$; define $f$ – is it all deterministic parts of RHS including deterministic part of the subgrid model?

- Line 343: Give a reference or explain "change of variables".

**References**

[1] Agarwal N, Kondrashov D, Dueben P, Ryzhov E, Berloff P. A Comparison of Data-Driven Approaches to Build Low-Dimensional Ocean Models. Journal of Advances in Modeling Earth Systems. 2021 Sep;13(9):e2021MS002537.

---

## Author Comment (AC1)

**Using Probabilistic Machine Learning to Better Model Temporal Patterns in Parameterizations: a case study with the Lorenz 96 model.**

**Author response**

We would like to thank both our reviewers for taking the time to provide such great in-depth comments and suggestions. We have made the appropriate changes and believe the manuscript is notably improved as a result. We summarise the key changes here and go into further detail below. From Reviewer 1, our most extensive change was to retrain the GAN and polynomial to allow a fairer comparison to the L96-RNN. We also improved the procedure for calculating KL divergences and edited the appropriate text to clarify this. From Reviewer 2 (Pavel Perezhogin), we used their suggestions to greatly improve the presentation of our model and the description of its loss functions, and also included a number of temporal autocorrelation results.

**Referee #1**

**Section 1**

**1.1 Missing context and clarity in manuscript setup**

The authors do a nice job of framing the issue that small-scale and unresolved processes are the greatest drivers of uncertainty in the cloud-climate feedback. However, given its importance to the premise of the manuscript, the authors need to give an explanation of what exactly "red noise" is rather than just saying it is important to parameterization. Likewise for "white noise" mentioned but not explained (and possibly blue noise as well). This will make the work more approachable to a more general audience. There are other examples of this throughout the manuscript but one more to point out is that, especially given it is the model of choice for the authors, the difference between traditional RNNs and LSTM should be explained more, and how specifically it is helpful in this framework as well as terms like "gated".

Thank you for pointing out areas where technical terms are not clarified. We agree that doing so would help the article's appeal to a more general audience. Hence, we have defined and given examples of red noise and white noise in lines 16-20, and also clarified the mentioned points on RNNs in lines 82-85.

We have included the additions to lines 16-20 here:

"[red noise] is characterized by a power spectral density which is inversely proportional to the square of the frequency. It is the type of noise produced by a random walk, where independent offsets are added to each step to generate the next step. This leads to strong autocorrelations in the time series. It contrasts a white noise process where each step is independent."

**1.2 clarity in Training procedure**

The authors do include some important information regarding the development of their model but more information is needed for reproducibility. The authors should clarify the process of hyper-parameter tuning that lead to their final hyper-parameter choices. What was tried and what wasn't? The authors should also clarify if they did a training/validation/test split or if they just do a training/validation split – and if the latter they should provide some justification for this decision. Additionally, did the authors experiment with different training/validation splits before settling on the final one, and if not a justification should be provided for the absence of this test.

We agree that extra details would help reproducibility and have now included them. We conducted a grid search for the hyper-parameters and this is now detailed in lines 203-205. We did a typical training/validation/test split, where the data in the training:validation set was in a ratio of 75:25 as now detailed in lines 197-200. We chose to use a classical training-validation split ratio and not to experiment on the exact split ratio as this would have required another validation set to assess this hyper-parameter of splitting ratio. By not doing this we had more data available for training.

I appreciate the authors details in Section 4.4 about computational costs between different models. But in terms of reproducibility, it would be helpful to include the amount of gpu/cpu/tpu hours required to train and tune these models.

We have included these at the end of Section 4.5.

**1.3 Climate valuation Metric lacks justification**

We believe that a lack of clarity in this section of our manuscript has led to confusion on our approach here. We have therefore extensively edited the appropriate section (4.2) and included details addressing specific points below.

I think the idea of approximating the true continuous distributions through a discrete one is likely appropriate here. But it needs proper justification and explanation (and citations). The authors are deriving the approach through "vector quantization" [2, 4], something they should introduce and explain. But through this vector quantization, the authors are not getting a true KL divergence, but rather estimating the lower bound of the true KL Divergence and their equation in 4.2 (not numbered) should be adjusted to reflect this for accuracy [1].

The idea of estimating an underlying continuous distribution's density with a discrete distribution based on a finite set of points is commonplace and referred to as histogram

density estimation. This is now detailed (with appropriate citations) in lines 227-229. The approach of "vector quantization" is a more general and sophisticated approach, with various techniques available for creating the discrete bins. We choose not to include this as the histogram density estimation method is far simpler and we do not believe it helps the reader to introduce the more general concept of vector quantization.

**The comments on KL divergence are addressed below.**

Because this is an estimation it requires additional justification. The principle (as though be included in this explanation) is that the more "bins" the more accurate the approximation of the lower bound of the KL Divergence. The authors need to justify the number of bins. More specifically they need to include either a graph or table (would recommend including this in the appendix rather than the main text) showing how the approximated KL Divergence changes as bin count is increased and at what bin count the approximated KL Divergence converges. This is the bin count the authors should use. As it stands now we cannot interpret any significance from the results because we don't know if the approximation of the KL Divergence they use is accurate or not.

Thank you for raising the point about the number of bins. The number of bins we chose made visualisation difficult and also was not the best representation of the underlying distribution. We have adjusted this accordingly. We explain in lines 229-233 that both too few bins and too many bins will lead to poor representations of the underlying distribution. Therefore, we use the Freedman-Diacanois rule (explained in lines 229-233) to choose the appropriate number of bins. Nevertheless, we also include, in Appendix B, KL divergences for a range of bin numbers to show our results still hold.

We note that the KL divergence calculated is indeed an approximation to the KL divergence of the underlying continuous distributions, but the approximation does not necessarily get more accurate as the number of bins increases, nor is the approximation a lower bound. Regarding the first point (accuracy as number of bins increases), we present an illustrative example: the KL divergence between two normal distributions,  $p \sim N(0,1^2)$  and  $q \sim N(0,2^2)$ , KL(q|p) = 0.807. It is finite. Now, consider if we were to sample from these distributions and represent them using histograms. If we used too many bins, there would be bins with data from q but not data from p, which would lead to an infinite KL divergence. Therefore, it is not true that the approximation becomes more accurate with more bins. The same example can be used for the second point (calculated KL divergence being a lower bound of the true KL divergence) - this is not the case as the calculated KL divergence could be infinite whereas the underlying distributions' KL divergence is finite. A similar example is given in lines 240-244. It is therefore not the case that the approximation is a lower bound.

Additionally, they need to confirm that Figures 3, 5, and 7 are built off of common bins. It would also be appropriate for the authors to acknowledge they are not the first to use this approach, it has been used in both supervised and unsupervised frameworks [1, 3]. Lastly, given that the KL Divergence is non0symmetric, there is an argument that the authors should symmetrize their approximated KL Divergences into Jensen-Shannon Divergences.

The figures are built using common bins. We now include the number of bins in the figure captions. We acknowledge that we are not the first to use such an approach and mention it

is commonplace (with appropriate citations) in lines 227-229. We agree it is helpful to explain why we use the KL divergence instead of the JS divergence, for example. We do this due to the KL divergence's equivalence to likelihood as is now noted in lines 235-238.

**Section 2**

(Line 25-27) [Add citation to paper with an idealized model.]

Many of these works do in fact use idealized models. For example, Bolton and Zanna, (2019) use an idealized ocean model, whilst Brenowitz and Bretherton (2018) use an idealized atmospheric model.

(Lines 31) [Add a reference to a section/figure that supports this claim.]

**Done.**

(Line 65) [Please be careful about calling any neural networks "deterministic" For most, the stochastic gradient descent and random seedings in initialization can/do add some variance that makes the label "deterministic" inaccurate.]

We clarify this point in line 69. Whilst the training of the neural networks is indeed stochastic in nature, we are referring to how the trained models are deterministic: for a given input you always get the same output.

(Lines 73-73) [Ciation needed for the claim about minmax loss.]

Done.

(Lines 104-105) [Is there a physical justification for this in terms of phenomena in the atmosphere? If so, would be helpful to mention that here.]

We have now added (lines 113-114) a justification in terms of error doubling times in the model and in the atmosphere.

(Line 105) [Word "good" is opinionated and vague.]

(Line 159) [Language is unscientific.]

Edited word choice for both cases above.

(161 and elsewhere) [Would suggest changing the term "truth data" to the more widely used "target data" to avoid confusion.]

We have kept this terminology but now defined it in line 192. We use "truth" data to refer to any data from the two-level L96 simulator. We chose not to use "target" data as that often

refers to data used to train models. And there are cases where "truth" data, coming from the L96 two-level simulator, is not "training" data (i.e. is not used for model training) — for example, we generate 50,000 MTU data from the two-level L96 simulator for climate evaluation. This is "truth" data but it is not used to train any models.

Another connotation of "target" is that this is the data we wish our models to create/simulate. But this is not the case: there are many possible valid trajectories which could be simulated given initial conditions only specified by X (and not Y). Therefore, there is not just one "target" which we wish for our models to generate.

(Line 163) [Why is the GAN only trained on F = 20? If the authors are trying to claim their model is superior, should it not be trained similarly to the RNN. Otherwise, the scope of the claim needs to be limited to moving beyond the specific paper the GAN was from rather than the approach itself.]

**Thank you for raising this point. You are right. We therefore retrained the GAN and polynomial model on the same data as the RNN.**

(Figure 2) [Need y axis label and title]

(Figure 3) [Remove ticks on the y-axis of plot b. Add titles.]

(Figures 3, 5, 7) [It is difficult to observe anything quantitative in these plots (as the authors themselves state in line 193). If the authors want to keep them, I would suggest moving to the Appendix.]

(Figure 4) [Add a title. Remove x ticks on a,b. Remove y ticks on b, d. This will allow the plots to be a bit bigger. Make the colorbar discrete to match the plots]

(Figure 5) [Remove y ticks from b. Add a title.).]

(Figure 7) [Remove y ticks from b. Add a title.).]

(Figure 5) [Remove y ticks from b, d and x ticks from a.c. Add a title. Set same y axis max on all four plots.).]

We have made changes to our plots according to these comments. The histograms are now clearer after using more bins so we have kept these figures in the main paper. We have also chosen to sometimes keep x/y ticks if we think they aid figure clarity.

(Line 209) [Explicitely define PCx as "Principle Component".]

Done.

(Lines 259-263) [Add citation.]

Done.

(Lines 270) [Reference the appropriate table.] (Lines 277) [Reference the appropriate table.]

These references have been included.

(Lines 310-312) [Would suggest including references to diffusion models or VAEs here.]

Lines 310-312 in the original preprint were discussing alternative ways to train the RNN which we had tried (i.e. adversarial training as opposed to likelihood). Therefore, we chose not to refer to VAEs/diffusion models as we did not think they helped the narrative. We have discussed the use of Transformers in lines 363-365, noting how they may better capture temporal patterns than RNNs.

**References**

Bolton, T. and Zanna, L.: Applications of deep learning to ocean data inference and subgrid parameterization, Journal of Advances in Modeling Earth Systems, 11, 376–399, 2019

Brenowitz, N. D. and Bretherton, C. S.: Prognostic validation of a neural network unified physics parameterization, Geophysical Research Letters, 45, 6289–6298, 2018.

**Referee #2 (Pavel Perezhogin)**

**Section 1**

**1.1 RNN model formulation**

The form of the model (Eq. 10-12) may be not familiar to general readers. Below I provide possible details that may be included to the text to improve the readability (up to the author's decision):

- The subgrid parameterization is split into the deterministic part (parameterized by mapping g $\theta$  )

and stochastic residual r which is parameterized by RNN.

• The RNN model is given in equations (11-12), and it attempts to predict a sequence of residuals at the next time step rk,t+1 given a sequence of rk,t.

• The major building block of the RNN is the mapping  $s\theta$ , which takes as an input the sequence of residuals rk,t and updates a hidden state lk,t recurrently.

• The output layer of the RNN gives a probabilistic prediction of the residual at the next time step rk,t+1 with a Gaussian distribution parameterized by b $\theta$  and z.

• Because sequences rk,t and rk,t+1 are related to each other, the stochastic prediction is fed back to the input of the RNN.

Also:

• It would be nice to give a reference to the exactly same form of the RNN model (11-12) if it is possible.

Thank you for these very helpful additions which definitely improve the clarity of our model description. We have included all of these in Section 3.1. We did make slight changes in

terminology, by referring to R as the stochastic part as opposed to the stochastic residual, since the notion of "residual" is now included in a later section (4.4).

**1.2 Training algorithm**

Line 156: If it is possible, give a reference for the training of RNNs with (similar) likelihood.

**We have included references to training RNNs with likelihood in lines 172-173 now.**

The presented description of the training algorithm and loss function (Line 156) are insufficient to reproduce the results. There are missing explanations that:

• The RNN model is trained given the time series of model state xt and the true subgrid forcing Ut (or coupling term).

• Be more explicit in "Where r and I are deterministic functions of x0, ..., xn derived from equations (10)–(12)", particularly:

- The residual is defined by  $r = Ut - g\theta$  (xt).

 – I is obtained from equation (12) which is solved given the known history of r and initial condition for I0

• An explicit expression for the loss function is required after changing log P r(r|I) by Gaussian probability density with log N (b $\theta$  (I),  $\sigma^2$ ) = const - 1/2 $\sigma^2$  (Ut - g $\theta$  (xt) - b $\theta$  (I))2. In my belief, the final loss involving most of the trainable mappings and parameters (g $\theta$ , b $\theta$ ,  $\sigma$ ), can be helpful to understand the model. For example, it is clear that both deterministic part of the model (g $\theta$ ) and model of stochastic residual (b $\theta$  (I)) are trained to jointly minimize the MSE of subgrid forcing prediction. This joint optimization clearly makes this model superior to the typical approach when first deterministic part is fitted, and then the residual is simulated independently, as in the Polynomial model. This interpretation of the loss may support the statement in Line 285 "For example, ... hardly suffered". I suggest to put the final loss and its interpretation in the Appendix.

Thanks for these insightful comments and the deep-dive into the training and loss formulation. We agree that the previous explanations did not aid reproducibility and hindered insight being gained about the training process. Therefore, we have reworked all of Section 3.2 to include these suggestions, with the explicit loss shown in equation (15).

**1.3 Temporal correlation**

The full paper is devoted to the temporal correlation, but there is no any plot of the lagged-autocorrelation function (ACF). Provide please ACF for: true subgrid forcing and simulated subgrid forcing (for Polynomial, GAN and RNN), and ACF for true and simulated residuals (r in case of RNN and h in case of Polynomial).

This was indeed a missing part of our results section. Figures 6 and 7 of the paper now show the ACF for our X variables and sub-grid forcing terms (U), respectively. We have also

included the ACF for the residuals - an important plot to check modelling assumptions - in Figure 11, and this indeed supports the usefulness of the RNN over a simple AR1 process as the L96-RNN's residuals decay quicker.

**1.4 Limitations**

Here I suggest topics to be clarified in the Discussion section.

The described approach to train RNN model requires computation of the likelihood of the trajectory (x1, x2, ..., xn), but not the residuals (or subgrid forcing), and it is not common. In the appendix, it is explained that for L96 system both likelihoods can be related to each other (A3):

 $\log Pr(xt|xt-1...x1) = \log P r(rt|rt-1...r1) - K \log(\Delta t) (1)$

It remains unclear if this trick is limited to scalar time series generated by L96. So, the discussion section can be improved with the following topic:

• Does the likelihood of the trajectory remain computationally tractable in the general case of GCM model (many correlated time series).

The following points are now discussed in lines 373-384 in the revised manuscript. The typical approach is indeed to maximise the likelihood of variables such as the sub-grid terms. For the L96, maximising the likelihood of the X trajectory is equivalent to maximising the likelihood of the trajectory of the hidden variables. However, in general, this may not be equivalent to maximising the likelihood of the X trajectory, and this is therefore undesirable. Nevertheless, it is typical to maximise the likelihood of the sub-grid terms as it is easier to do so. We note that in future work we plan to look at how to better train models where you cannot maximise the likelihood of the X trajectory (or the likelihood of a sequence of variables whose likelihood is proportional to that of the X trajectory).

Also:

• In spite of the joint optimization of deterministic part  $g\theta$  and stochastic residuals r, equations for stochastic residuals (11-12) are not informed with the solution xt. So, the presented model is not conditional (unlike the GAN model). Is it possible to make it conditional?

We have included in lines 367-372 a note on this, with the explanation that we could indeed make the model conditional if it was desired. We chose not to do so to allow for a cleaner comparison against red-noise.

Section 2

• Line 76, "eliminating the need for update functions to be manually specified". Make a reference to  $\beta$  or Eq. (3).

• Consider citation of stochastic LSTM model applied to geophysical data [1].

**Both of the above have been updated.**

• Line 4: "physically-informed" and Line 289 "physical features", Line 297 "mainly those which did not leverage physical structure" what is meant in these cases?

We are referring to the keeping of the physical structure from terms like advection in the model (equations (10)-(12)), as opposed to just training a NN to learn the complete mapping itself. We clarify this in line 354, where we discuss how "conventional parameterizations are augmented with a stochastic term learnt using ML". We have also edited lines 355-357. We have removed the term "physically-informed" from the Abstract to improve clarity.

• Line 9 and in other places "probabilistic metric of hold-out likelihood". Please specify that it is offline metric (computed on the trajectory of the true system).

We have now included this, along with further detail on how this does not guarantee good online performance, in lines 336-339.

• Line 41: "invented hidden variables", what does "invented" mean.

We mean that these are created by the modeller, as opposed to being a physical state variable for example. We have now changed the wording in line 44.

• Equation 5: define  $\varphi$  and  $\sigma$  (as lag-one correlation and standard deviation of time series)

**Done.**

• Line 101: in L96 equation for X, check j, k indexes in coupling term.

Thanks for noting this. There was an error in the indices and we have now corrected it.

• Equation (7): change  $\lambda(Xt/2)$  to  $1/2\lambda(Xt)$

**Done.**

• Line 168: Provide the number of training points in the epoch.

We have now included the number of training samples (3,999,992) in the training set. These are the number of points that are used per training epoch.

• Line 181: I do not think that  $t = tinit + \tau$  helps the reader, and most likely wrong.

We agree it probably does not help the reader, so have edited the notation.

• Line 182: Why Xm,n(t) does not have index "sample", but Xsample m has. They should be the same, I think.

Thanks, we have corrected this.

• Line 209: Give a short definition of "in a phase quadrature"

In line 256 we provide an explanation of what we meant (being offset in phase by  $\pi/2$  radians). We have removed the term "in phase quadrature" as we believe it no longer helps the reader.

• Line 212: Is ||[P C1, P C2]|| = √P C12 + P C22.

Yes. We now define this in line 259.

· Line 230: Check "The to simulate"

**Corrected.**

• Line 256: Clarify "despite the polynomial exploding"

**Done in lines 279-280.**

· Lines 259-262: Reformulate the whole paragraph.

**Done.**

• Line 289: "the the"

**Corrected.**

• Figure 9: "Model which allows to learning from high-resolution data" is not clear. The word "learning" should be specified. As I understand, it is proposed to simulate unresolved variables with RNN.

The proposition is to learn using the high-resolution variables, but at simulation time not require them being simulated when simulating the low-resolution variables. The figure caption (Figure 12 now) has been updated.

• Line 293: "There are more sophisticated ways to model the system": reformulate.

**Done.**

• Line 310: Probably better to say "GAN loss function to train RNN".

Have made edits to say something similar to this, highlighting how the adversarial approach to training is that which is used for GANs (lines 395-397).

• Check Discussion and Conclusion: the main results of the paper are not stated clearly, i.e. there is no summary.

**Updated. A summary is now provided in lines 405-411.**

• Equation (A2): Clear distinction between xt and Xt will help; define the notation for parentheses (x|y); define f – is it all deterministic parts of RHS including deterministic part of the subgrid model?

We agree with needing to clarify the distinction between upper and lower case. We have now done this in lines 153-157. We now define f in equation A3.

• Line 343: Give a reference or explain "change of variables".

We have written a new section, A4, to explain this.